# The meningeal transcriptional response to traumatic brain injury and aging

Ashley C Bolte[1,2,3,4]*[†], Daniel A Shapiro[1][†], Arun B Dutta[3,5], Wei Feng Ma[3,6], Katherine R Bruch[1], Michael A Kovacs[1,2,3,4], Ana Royo Marco[1,2], Hannah E Ennerfelt[1], John R Lukens[1,3,4]*

[1]Department of Neuroscience, Center for Brain Immunology and Glia (BIG), University of Virginia School of Medicine, Charlottesville, United States; [2]Department of Microbiology, Immunology and Cancer Biology, University of Virginia School of Medicine, Charlottesville, United States; [3]Medical Scientist Training Program, University of Virginia School of Medicine, Charlottesville, United States; [4]Immunology Training Program, University of Virginia School of Medicine, Charlottesville, United States; [5]Department of Biochemistry and Molecular Genetics, University of Virginia School of Medicine, Charlottesville, United States; [6]Center for Public Health Genomics, University of Virginia School of Medicine, Charlottesville, United States

*For correspondence:
aco5uv@virginia.edu (ACB);
jrl7n@virginia.edu (JRL)

[†]These authors contributed equally to this work

Competing interest: The authors declare that no competing interests exist.

**Abstract** Emerging evidence suggests that the meningeal compartment plays instrumental roles in various neurological disorders, however, we still lack fundamental knowledge about meningeal biology. Here, we utilized high-throughput RNA sequencing (RNA-seq) techniques to investigate the transcriptional response of the meninges to traumatic brain injury (TBI) and aging in the sub-acute and chronic time frames. Using single-cell RNA sequencing (scRNA-seq), we first explored how mild TBI affects the cellular and transcriptional landscape in the meninges in young mice at one-week post-injury. Then, using bulk RNA-seq, we assessed the differential long-term outcomes between young and aged mice following TBI. In our scRNA-seq studies, we highlight injury-related changes in differential gene expression seen in major meningeal cell populations including macrophages, fibroblasts, and adaptive immune cells. We found that TBI leads to an upregulation of type I interferon (IFN) signature genes in macrophages and a controlled upregulation of inflammatory-related genes in the fibroblast and adaptive immune cell populations. For reasons that remain poorly understood, even mild injuries in the elderly can lead to cognitive decline and devastating neuropathology. To better understand the differential outcomes between the young and the elderly following brain injury, we performed bulk RNA-seq on young and aged meninges 1.5 months after TBI. Notably, we found that aging alone induced upregulation of meningeal genes involved in antibody production by B cells and type I IFN signaling. Following injury, the meningeal transcriptome had largely returned to its pre-injury signature in young mice. In stark contrast, aged TBI mice still exhibited upregulation of immune-related genes and downregulation of genes involved in extracellular matrix remodeling. Overall, these findings illustrate the dynamic transcriptional response of the meninges to mild head trauma in youth and aging.

## Editor's evaluation

The authors provide single RNA-seq analysis of traumatic brain injury (TBI) that particularly addresses the question of why older individuals may have poor recovery. Compelling complementary and high-end approaches are taken to demonstrate the long-lasting effects that TBI drives in the brain. This important manuscript will be of interest to readers in the field(s) of neuroimmunology, aging, and traumatic brain injury.

## Introduction

Traumatic brain injury (TBI) affects millions of people each year and can result in devastating long-term outcomes (*Marin et al., 2017*; *Roozenbeek et al., 2013*; *Smith et al., 2013*; *McKee et al., 2009*; *Faul and Coronado, 2015*; *Selassie et al., 2008*; *Kang and Lin, 2012*; *Fann et al., 2018*; *Frost et al., 2013*; *Alexis et al., 2014*). While TBI affects individuals of all ages, the elderly experience more severe consequences than younger individuals with similar injury severity (*Susman et al., 2002*). The reason for this differential age-related response to brain injury is not fully understood. Multiple findings have indicated that prolonged activation of the immune system following TBI may contribute to some of the negative TBI-associated sequelae (*Ertürk et al., 2016*; *Winston et al., 2016*; *Corps et al., 2015*; *Johnson et al., 2013*; *Schimmel et al., 2017*; *Chou et al., 2018*; *McKee and Lukens, 2016*; *Witcher et al., 2021*). Interestingly, several studies point to differences in the immune response in elderly individuals that may contribute to more severe consequences following injury (*Chou et al., 2018*; *Morganti et al., 2016*; *Ritzel et al., 2018*; *Webster et al., 2015*; *Androvic et al., 2020*; *Ritzel et al., 2019*; *Kumar et al., 2013*; *Krukowski et al., 2018*). However, our understanding of the disparate CNS responses between elderly and young individuals following TBI is still in its infancy.

Recent findings have implicated the meninges, a tri-layered tissue that resides between the brain parenchyma and skull, as an early responder to TBI and as a pivotal contributor to the CNS immune response following injury (*Russo et al., 2018*; *Roth et al., 2014*). Meningeal enhancement with post-contrast fluid attenuated inversion magnetic resonance imaging (MRI) can be seen in 50% of patients with mild TBIs and no apparent parenchymal damage (*Roth et al., 2014*). This enhancement has been shown to occur within minutes of injury (*Turtzo et al., 2020*). Moreover, many individuals who experienced mild TBIs still exhibited extravasation of contrast into the sub-arachnoid space, indicating that the blood-brain-barrier was compromised (*Turtzo et al., 2020*). While most patients experienced resolution in meningeal enhancement 19 days after injury, about 15% had persistent enhancement three months post-injury, indicating that some patients experienced prolonged periods without complete meningeal repair following mild TBI (*Russo et al., 2018*). These protracted periods of meningeal enhancement likely represent ongoing inflammation within the compartment, yet the different cellular and molecular components that drive this inflammation have not been fully investigated.

The meningeal response to brain injury can be divided into several phases: acute, sub-acute, and chronic (*Roth et al., 2014*). Initial studies of the acute phase response after a mild TBI detail a meningeal response that consists of rapid meningeal cell death due to vascular leakage and reactive oxygen species release, which results in secondary parenchymal damage within the first several hours of injury (*Russo et al., 2018*; *Roth et al., 2014*). The initial injury is followed by meningeal neutrophil swarming (present within an hour of injury) that is essential for regeneration of the initially damaged glial limitans (*Roth et al., 2014*). Disrupted meningeal vasculature is then repaired during the week following injury by non-classical monocytes (*Russo et al., 2018*). While these acute meningeal responses have been investigated, much less is understood about how brain injury shapes the meningeal environment more chronically, and if this response is affected by aging. Furthermore, it is unknown whether chronic meningeal changes following brain injury in aged individuals can contribute to neurodegenerative processes.

In addition to housing lymphatic vessels that drain molecules and cells to peripheral lymph nodes (*Aspelund et al., 2015*; *Louveau et al., 2015*), the meninges also contain a full array of innate and adaptive immune cells that are in constant communication with neurons and glia (*Alves de Lima et al., 2020b*; *Alves de Lima et al., 2020a*; *Rustenhoven et al., 2021*). In homeostasis, cytokine signaling from meningeal immune cells has been shown to be critical for shaping cognition (*Alves de Lima et al., 2020a*; *Filiano et al., 2016*; *Derecki et al., 2010*; *Ribeiro et al., 2019*). For instance, IFN-γ is important in maintaining social behavior networks, whereas IL-4 production by meningeal T cells has been shown to influence learning and memory (*Filiano et al., 2016*; *Derecki et al., 2010*). Recent studies also suggest that IL-17a secretion by γδ T cells in the meninges can impact anxiety-like behaviors and memory (*Alves de Lima et al., 2020a*; *Ribeiro et al., 2019*). Of particular relevance, recent work has shown that an age-related decline of CCR7 expression by meningeal T cells may contribute to cognitive impairment, brain inflammation, and neurodegenerative disease (*Da Mesquita et al., 2021*). In addition to meningeal T cell production of cytokines, it is known that immune cells within the cerebrospinal fluid (CSF) in the subarachnoid space can also produce signaling molecules and interact with brain-derived products and antigens (*Gate et al., 2020*; *Lepennetier et al., 2019*). Brain

interstitial fluid (ISF) and CSF intermix in the subarachnoid space and both recirculate throughout the brain via the glymphatic system and drain through the meningeal lymphatic network to the periphery (*Aspelund et al., 2015*; *Louveau et al., 2015*; *Plog et al., 2015*; *Ringstad and Eide, 2020*; *Louveau et al., 2017*; *Goodman and Iliff, 2020*; *Antila et al., 2017*; *Iliff et al., 2013*; *Iliff et al., 2014*; *Iliff et al., 2012*; *Jessen et al., 2015*; *Peng et al., 2016*). This system provides meningeal cells and cells within the CSF with access to brain antigens and proteins. Despite mounting evidence demonstrating that meningeal cells can impact various aspects of neurobiology, we still lack a complete picture of how the meninges respond to processes that have been broadly linked to neurological disease, such as brain injury and aging. Likewise, little is known in regards to how aging impacts meningeal biology both under steady-state conditions and in response to TBI.

Here, we investigated how the meningeal transcriptional environment is altered following TBI and in aging utilizing high-throughput sequencing techniques, namely single-cell RNA sequencing (scRNA-seq) and bulk RNA sequencing (bulk RNA-seq). We focused on sub-acute (one week post-TBI) and chronic (1.5 months post-TBI) time points after brain injury to better understand how the cellular makeup and gene expression profiles in the meninges change with time and with age. We found that the heterogeneous cellular makeup of the meninges was altered one week following TBI with an increase in the frequency of macrophages and fibroblasts. Moreover, we further showed that the meningeal transcriptional environment was significantly altered in aging, including a broad upregulation of genes involved in antibody production and type I interferon (IFN) signaling. When examining the genes that were differentially expressed in aged mice as compared to young mice 1.5 months following TBI, we found that there was downregulation of genes important for extracellular matrix remodeling and collagen production, and an overall activation of immune system-related genes. This prolonged activation of the immune system was unique to the aged TBI mice, as the young mice exhibited few alterations in the meningeal transcriptome 1.5 months following injury. In order to identify subacute transcriptional changes that may persist chronically after injury in aging, we identified the shared differentially expressed genes between the bulk RNA-seq and scRNA-seq datasets. We found genes that are critical for maintaining an immune response that were initially upregulated one week after injury in young mice, remained upregulated in the chronic setting 1.5 months after injury. Genes important for connective tissue maintenance and wound healing that were downregulated one week after injury, remained downregulated 1.5 months after injury. These findings help reveal chronic dysregulated transcriptional response patterns seen in aging after brain injury. Overall, this study highlights the dynamic nature of the meningeal transcriptome in response to TBI and aging, and sheds light on some of the differences between young and aged individuals in responding to brain injury.

## Results
### Mild TBI incites alterations in the cellular composition of the meninges

To gain insights into how TBI impacts meningeal biology, we subjected mice to a mild closed-skull injury and then performed scRNA-seq on the meninges one week post-injury (*Figure 1a*). In this model of mild TBI, mice received a single hit to the right inferior temporal and frontal lobes using a stereotaxic electromagnetic impactor (*Figure 1—figure supplement 1a*; *Bolte et al., 2020*). Of note, we have previously shown that head injury in this model does not result in appreciable alterations in balance, motor coordination, reflex, and alertness (*Bolte et al., 2020*). Consistent with the mild nature of this TBI model, we also do not observe any appreciable differences in CD31 blood vasculature staining at 24 hr following head trauma (*Figure 1—figure supplement 1b and c*). Moreover, we only detect modest increases in gliosis (Iba1 and GFAP staining) (*Figure 1—figure supplement 1d, e and f*) and MHCII + staining in the meninges at 24 hr post-TBI (*Figure 1—figure supplement 1g, h and i*).

For all of the sequencing studies in this paper, we strategically chose to isolate only the dorsal meningeal tissue, as this region of the meninges does not include the tissue affected by the direct injury site. Therefore, the sequencing data generated from these studies should better reflect the global meningeal changes that result from a localized injury site rather than the tissue damage and response at the immediate injury site. Joint clustering of both the Sham and TBI meninges revealed 21 unique cell populations including endothelial cells, fibroblasts, Schwann cells, and ciliated ependymal cells from the pia (*Figure 1b and c*, *Table 1*). Additionally, the meninges contained a full repertoire of immune cells including macrophages, B cells, T cells, NK cells, dendritic cells, plasmacytoid

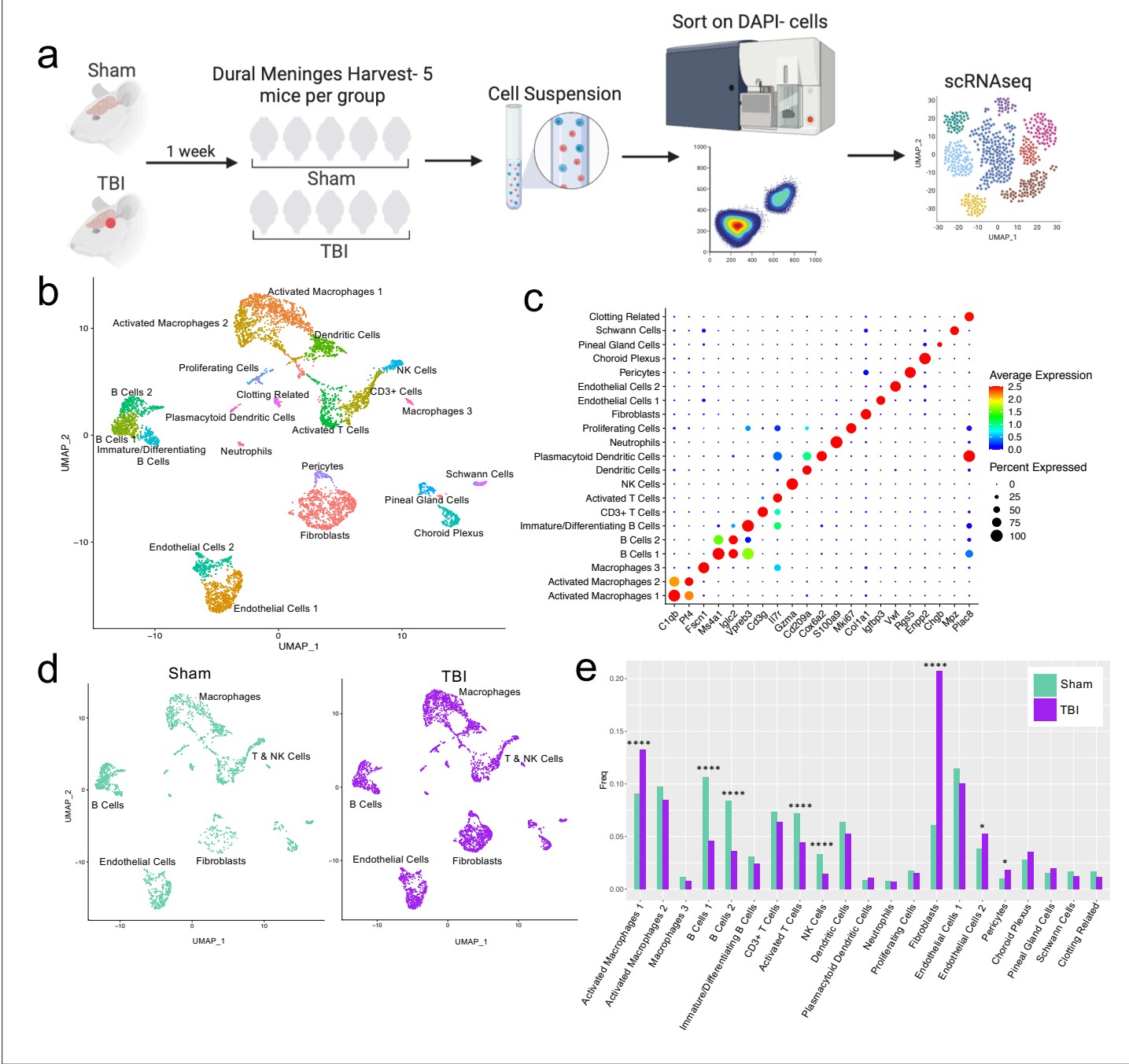

**Figure 1.** Alterations in the composition of meningeal cell populations following brain injury. Male C57BL/6 J wild-type (WT) mice at 10 weeks of age were subjected to a mild closed-skull injury above the right inferior temporal lobe or Sham procedure. One week later, the meninges from 5 mice per group were harvested, pooled, and processed for scRNA-seq. (**a**) Schematic of scRNA-seq protocol. (**b**) Uniform Manifold Approximation and Projection (UMAP) representation of the cell populations present in the meninges where both Sham and TBI groups are included. Colors are randomly assigned to each cell population. (**c**) Dot plot representation of cluster defining genes for each cell population, where each gene represents the most significant cluster-defining marker for each population. The color and size of each dot represents the average expression and percent of cells expressing each gene, respectively. (**d**) UMAP representations of the cell populations present in the meninges separated by Sham (sage) and TBI (purple). (**e**) Bar graph depicting frequencies of cell populations in Sham vs. TBI samples. Graphs were calculated using Seurat by normalizing the dataset, finding the variable features of the dataset, scaling the data, and reducing the dimensionality. Each data point in a UMAP plot represents a cell. p Values were calculated using a two sample z-test. ****p<0.0001, *p<.05, bar chart pairs without * were not statistically significant. Exact statistics are provided in the source data file.

The online version of this article includes the following source data and figure supplement(s) for figure 1:

*Figure 1 continued on next page*

*Figure 1 continued*

**Source data 1.** Cluster-defining genes for single cell populations.

**Source data 2.** Frequency, cell count, and p-value for each single cell population.

**Figure supplement 1.** Initial brain and meningeal response following TBI.

**Figure supplement 1—source data 1.** Table depicting the percent area of CD31 + immunofluorescence in the meninges 24 hr after TBI.

**Figure supplement 1—source data 2.** Tables depicting the percent area of GFAP + and Iba1 + immunofluorescence in the brain 24 hr after TBI.

**Figure supplement 1—source data 3.** Table depicting the percent area of MHCII + immunofluorescence in the meninges 24 hr after TBI.

**Figure supplement 2.** Stress and processing related genes after single cell RNA-sequencing.

dendritic cells, and neutrophils (*Figure 1b and c*, *Table 1*). Other cell populations were less well-defined and included cells expressing genes important for clotting and proliferating cells (*Figure 1b and c*, *Table 1*). When separated out by Sham and TBI treatments, all 21 populations were still present in both groups (*Figure 1d*, *Table 1*), however the frequencies were varied (*Figure 1e*). Following brain injury, there was a higher frequency of one group of macrophages which we denoted as 'Activated Macrophages 1' as they exhibit high expression of complement-related genes (*Figure 1c and e*). Moreover, the frequency of fibroblasts was substantially increased following head trauma (*Figure 1e*). While there was a reduction in frequency of some other cell types, namely the B cell populations, it is unclear whether this was relative to the expansion of the other subsets or an actual decrease in number (*Figure 1e*). In order to ensure the short digestion and processing steps of the sample

**Table 1.** Counts of each cell population separated by Sham and TBI.

Male WT mice at 10 weeks of age received a TBI or Sham procedure. One week later, the meninges from 5 mice per group were harvested, pooled, and processed for scRNA-seq. The cell counts for each cell population are shown after data processing.

| Cell Type | Cell Counts Sham | Cell Counts TBI |
|---|---|---|
| Activated Macs 1 | 185 | 499 |
| Activated Macs 2 | 199 | 319 |
| Macs 3 | 24 | 31 |
| B Cells 1 | 217 | 172 |
| B Cells 2 | 172 | 137 |
| Immature/Diff B Cells | 63 | 92 |
| CD3+T Cells | 150 | 240 |
| Activated T Cells | 148 | 169 |
| NK Cells | 68 | 55 |
| Dendritic Cells | 131 | 200 |
| Plasmacytoid Dendritic Cells | 18 | 42 |
| Neutrophils | 16 | 26 |
| Proliferating Cells | 36 | 58 |
| Fibroblasts | 124 | 781 |
| Endothelial Cells 1 | 235 | 380 |
| Endothelial Cells 2 | 78 | 199 |
| Pericytes | 20 | 69 |
| Choroid Plexus | 58 | 135 |
| Pineal Gland Cells | 31 | 75 |
| Schwann Cells | 35 | 47 |
| Clotting Related | 34 | 43 |

preparation did not result in significant upregulation of stress-related genes in both Sham and TBI samples, we examined a collection of genes that have been known to be upregulated after tissue processing and in stress-related conditions (*Haimon et al., 2018*; *Marsh et al., 2020*; *Van Hove et al., 2019*; *Figure 1—figure supplement 2*). Very few sequenced cells expressed these genes and there were not substantial differences between the TBI or Sham experimental groups, suggesting minor contributions of processing on gene expression and similar effects across experimental groups (*Figure 1—figure supplement 2*). Overall, these data highlight the heterogeneous nature of the meningeal tissue and also demonstrate that the frequencies of macrophage and fibroblast populations are increased 1 week post-TBI.

## Effects of mild TBI on the meningeal macrophage transcriptome

Given our data demonstrating an appreciable expansion of the meningeal macrophage population following injury (*Figure 1e*), as well as emerging data suggesting instrumental roles for these cells in TBI pathogenesis (*Russo et al., 2018*; *Roth et al., 2014*), we decided to focus first on the response of meningeal macrophages to head trauma. Differential gene expression analysis of the 'Activated Macrophage' populations (Activated Macrophages 1 & 2) revealed 321 upregulated genes and 369 downregulated genes following head injury when using a false discovery rate of <0.1 (*Figure 2a*). When we performed a network analysis on the significantly upregulated genes from these populations, we found an enrichment of pathways related to immune system activation (*Figure 2b*). Upregulated genes in the activated population included those important for cytokine secretion, immune cell differentiation, motility, and chemotaxis (*Figure 2b*). Furthermore, the most highly enriched gene ontology (GO) biological processes modulated in response to head trauma were found to be related to immune system activation and the stress response (*Figure 2c*). We also noticed that some of the most significantly upregulated genes contributing to the immune-related GO terms were important for the type I IFN response including *Ifnar1, Ifi203, Irf2bp2,* and *Irf5*, amongst others (*Figure 2d*). We next examined type I IFN gene expression by qPCR in whole meningeal samples one week after injury. We observed elevated expression of genes including *Ifnar1*, *Irf5*, *Ifnb1*, and *Ifi203*, highlighting the strong type I IFN gene signature after TBI that is likely driven by meningeal macrophages (*Figure 2e–h*). Interestingly, recent studies suggest that elevated type I IFN signaling in the brain parenchyma is a driver of detrimental outcomes in TBI pathogenesis (*Karve et al., 2016*; *Barrett et al., 2020*). Taken together, these findings suggest that meningeal macrophages upregulate inflammation-related genes one week following brain injury and may contribute to the type I IFN signature that is seen following TBI.

Previous findings also suggest that there are several subtypes of meningeal macrophages that respond to TBI (*Russo et al., 2018*). Therefore, we decided to look more closely at the subpopulations within the original macrophage clusters. We re-clustered the three macrophage populations (Activated Macrophages 1 & 2, and Macrophages 3) combined from Sham and TBI meninges, which yielded six different meningeal macrophage clusters (*Figure 2i*). The largest population of macrophages expressed high levels of ferritin ('Ferritin Expressing') (*Figure 2i*). The top two cluster-defining genes within the 'Ferritin Expressing' macrophages were ferritin light chain (Ftl1) and ferritin heavy chain (Fth1) (*Figure 2i*). There were two additional populations, deemed 'Anti-Inflammatory' and 'Resolution Phase' macrophages, that appeared to be alternatively activated, anti-inflammatory macrophages that are likely implicated in the healing response following injury. The top cluster-defining gene in the 'Anti-Inflammatory' macrophage cluster was *Mrc1* (also referred to as CD206), which is known to be present on macrophages that play a role in the healing response after TBI (*Russo et al., 2018*). Other highly significant subcluster-defining genes in the 'Anti-Inflammatory' macrophage population included *Stab1, Nrros,* and *Dab2*, which are known to be expressed on healing macrophages, and are important for repressing reactive oxygen species and limiting type I IFN responses (*Park et al., 2009*; *Noubade et al., 2014*; *Hung et al., 2016*; *Figure 2—figure supplement 1a*). 'Resolution Phase' macrophages do not fall into either the M1 classically activated or M2 alternatively activated macrophage categories and are believed to play a regulatory role following an inflammatory event (*Stables et al., 2011*). They tend to be enriched for antigen presenting genes, chemokine genes, and proliferation-related genes (*Stables et al., 2011*). Indeed, the meningeal macrophages in the 'Resolution Phase' cluster were defined by their expression of antigen presentation-related genes (*H2-Eb1, H2-Ab1, H2-Aa, Cd74, Ctss*) and anti-inflammatory genes such as *Lair1*, an inhibitory receptor that prevents over-activation of cytokine production (*Meyaard et al., 1997*; *Figure 2—figure supplement*

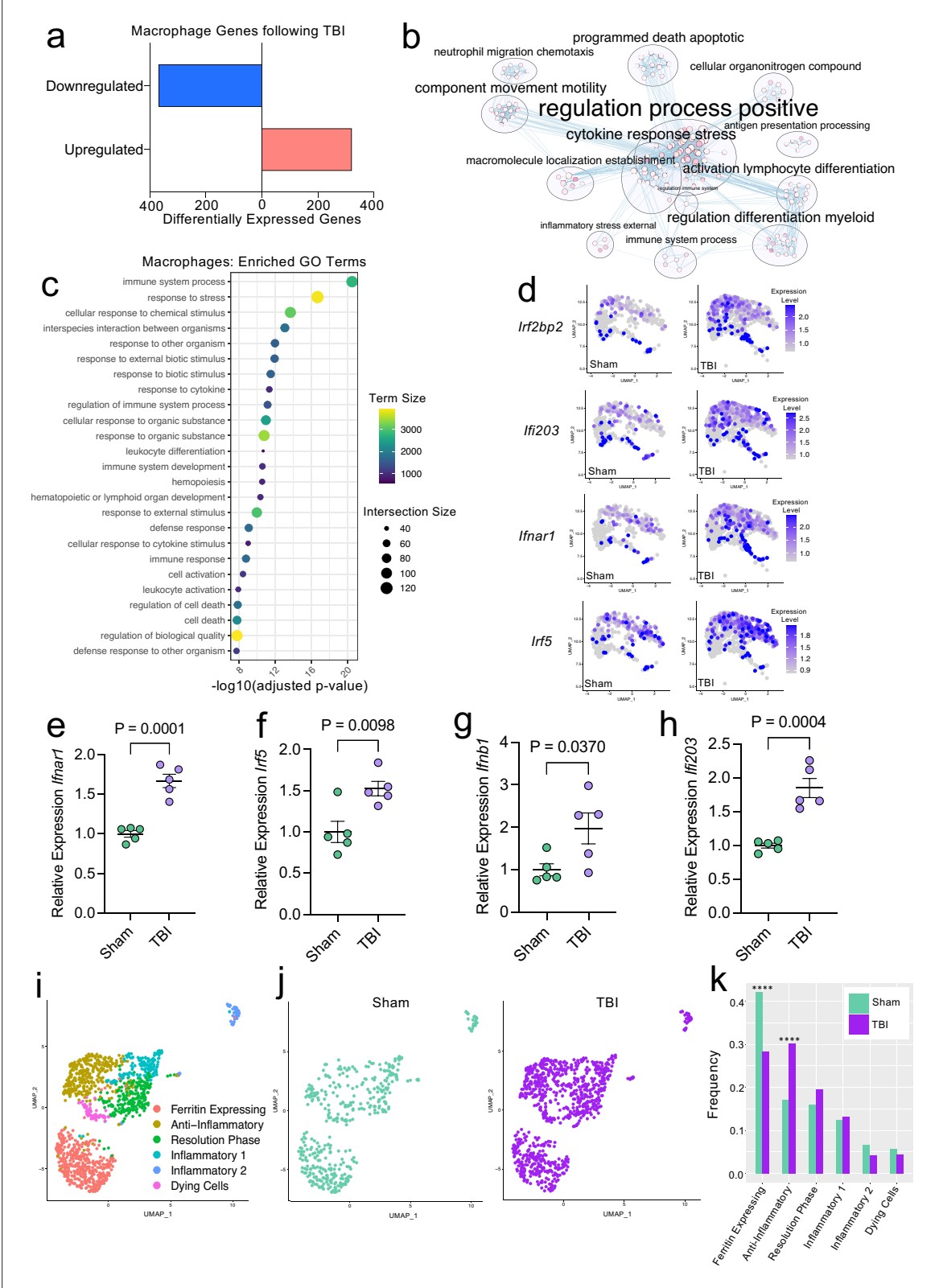

**Figure 2.** Transcriptional response of meningeal macrophages to mild TBI. Male WT mice at 10 weeks of age received a TBI or Sham procedure. One week later, the meninges from 5 mice per group were harvested, pooled, and processed for scRNA-seq. (**a**) Quantification of the number of upregulated and downregulated macrophage genes following injury (FDR < 0.1). (**b**) Network analysis of significantly upregulated genes in meningeal macrophages following injury. Text size is proportional to the number of genes enriched in that cluster. Node size is roughly proportional to the number of GO terms

*Figure 2 continued on next page*

*Figure 2 continued*

in that cluster (node size was manually adjusted so may not be exactly proportional to GO terms included). Dot size is proportional to the number of genes contributing to each GO term. Dot color is proportional to p-value, where colors closer to white have lower p-values. Connecting lines represent GO terms with shared genes, more lines represents a higher number of shared genes between nodes. (**c**) Dot plot showing the 25 most enriched GO terms with significantly upregulated genes following TBI in the meningeal macrophage population. The color and size of each dot represents the size of the GO term and the number of upregulated genes that contribute to each term, respectively. (**d**) Feature plots depicting several significantly upregulated genes following injury (FDR < 0.1). The color of each data point represents the expression level of the indicated gene within that cell. Quantitative PCR relative expression of (**e**) *Ifnar1*, (**f**) *Irf5*, (**g**) *Ifnb1*, and (**h**) *Ifi203* within the dural meninges one week after TBI (Sham n=5, TBI n=5, rep = 1). (**i**) UMAP representation showing re-clustering of the meningeal macrophage populations. (**j**) UMAP representation of the macrophages present in the meninges separated by Sham (sage) and TBI (purple). (**k**) Frequencies of meningeal macrophage populations in Sham vs. TBI samples represented as a gradient bar chart. Graphs were calculated using Seurat by normalizing the dataset, finding the variable features of the dataset, scaling the data, and reducing the dimensionality. Differential gene expression was calculated using the ZINB-WaVE function for zero-enriched datasets and DESeq2. Each data point in a UMAP plot represents a cell. Error bars in (**e–h**) depict mean ±s.e.m. p Values for (**e–h**) were calculated using unpaired two-sample students t-tests and p values for (**k**) were calculated using a two sample z-test. ****p<0.0001. Bar chart pairings without * were not statistically significant, exact statistics are provided in the source data file. FDR; false discovery rate.

The online version of this article includes the following source data and figure supplement(s) for figure 2:

**Source data 1.** Number of up- and downregulated macrophage genes 1 week post-TBI.

**Source data 2.** Raw data showing enriched GO-terms and contributory differentially expressed macrophage genes.

**Source data 3.** Raw data for the differential expression analysis in the macrophage population 1 week post-TBI.

**Source data 4.** Relative expression of interferon-related genes by qPCR in whole meninges 1 week post-TBI.

**Source data 5.** Cluster-defining genes for macrophage populations.

**Source data 6.** Frequency, cell count, and p-value for each macrophage population.

**Figure supplement 1.** Cluster-defining genes for macrophage subpopulations.

**Figure supplement 2.** Ligand-target interactions between macrophages and other meningeal cells.

**Figure supplement 2—source data 1.** List of top macrophage ligands likely to influence the gene expression patterns of the other major cell populations in the scRNA-seq dataset.

*1b*). In contrast to these 'Anti-Inflammatory' and 'Resolution Phase' clusters, the final two macrophage populations exhibited gene signatures more commonly associated with inflammatory macrophages (*Figure 2i*). The 'Inflammatory 1' macrophage cluster was defined by its differential expression of *Ccr2* and adhesion molecules such as *Alcam* and *Lgals3* (*Figure 2—figure supplement 1c*). The second inflammatory macrophage subset 'Inflammatory 2' was defined by its expression of genes important for chemotaxis including *Ccr7*, *Ccl22*, and *Ccl5* (*Kwiecień et al., 2019*; *Martinez and Gordon, 2014*; *Figure 2—figure supplement 1d*).

To determine how injury affected these distinct meningeal macrophage populations, we separated the cells into Sham and TBI groups and examined their frequencies (*Figure 2j and k*). Interestingly, there was an overall relative increase in the 'Anti-Inflammatory' and 'Resolution Phase' macrophages in the TBI group, indicating that one week after injury, the response of the meningeal macrophages appears to shift towards wound healing and inflammation resolution (*Figure 2j and k*). There was also a relative reduction in the 'Ferritin Expressing' macrophages following injury (*Figure 2j and k*). Overall, these findings demonstrate that although the macrophages collectively upregulated genes essential for inflammation following TBI, the frequencies of 'Resolution Phase' and 'Anti-Inflammatory' macrophages also increased and may play a role in the wound healing process.

## Mild TBI induces the upregulation of neurological disease-associated genes in meningeal fibroblasts

Next, we wanted to further investigate the 'Fibroblast' population, as it was expanded 1 week following injury in the single cell dataset (*Figure 1d and e*). After head injury, the 'Fibroblast' population exhibited 368 activated genes and 320 repressed genes (*Figure 3a*). There were several patterns in the activated genes following injury, including genes important for collagen production and extracellular matrix remodeling (*Nisch, Ppib, Pmepa1, Ddr2*) and genes critical for cell motility and growth (*Ptprs, Pfn1, Cd302, Tpm3*) (*Figure 3b*). To validate these sequencing findings, we utilized immunohistochemical staining for collagen, which is produced by fibroblasts, in meningeal whole mounts. Consistent with the sequencing results, we observed a significant increase in collagen density 1 week

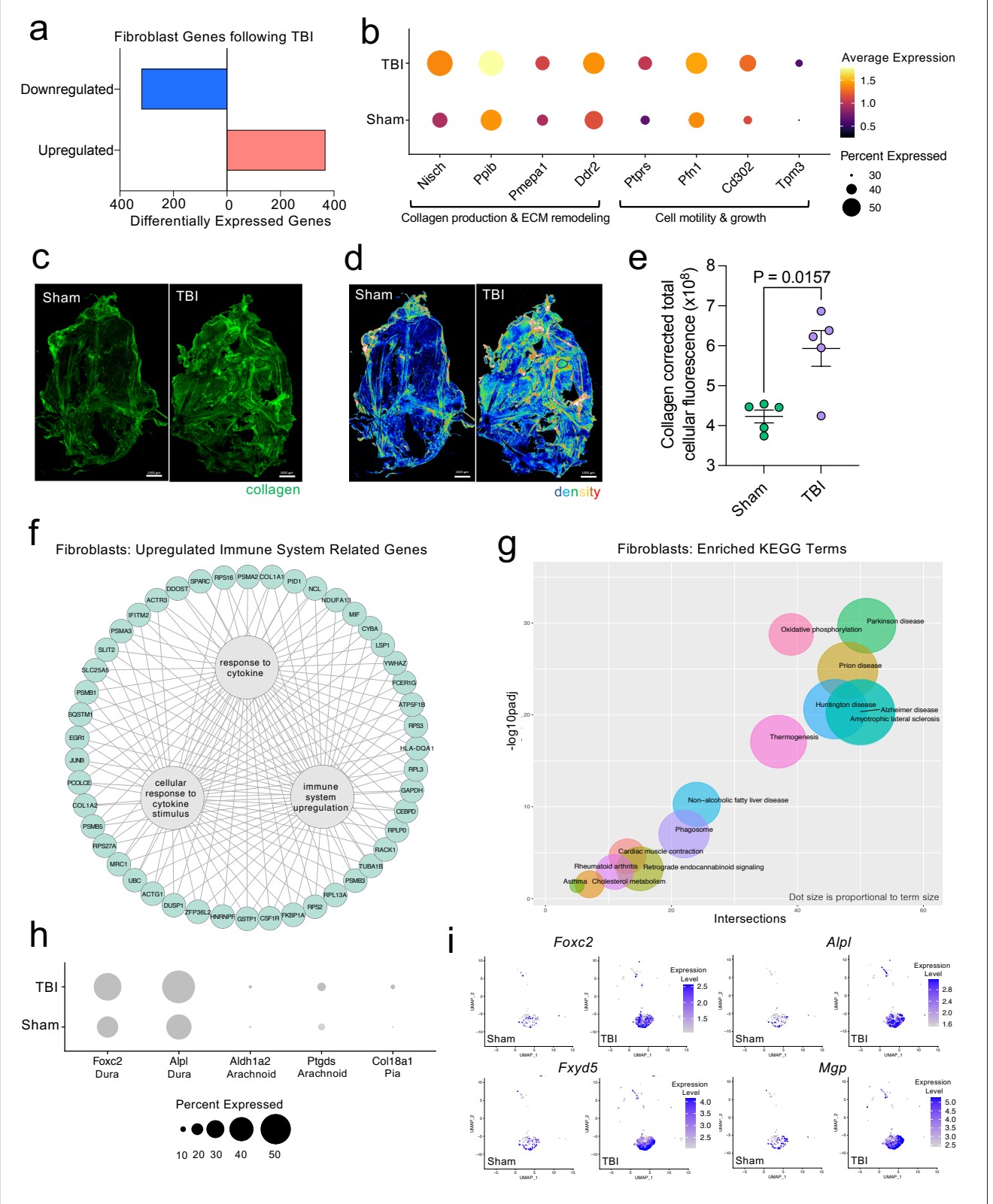

**Figure 3.** Dural fibroblasts express genes involved in tissue remodeling, cell migration, and immune activation in TBI. Male WT mice at 10 weeks of age received a TBI or Sham procedure. One week later, the meninges from 5 mice per group were harvested, pooled, and processed for scRNA-seq. (a) Quantification of the number of upregulated and downregulated fibroblast genes following injury (FDR < 0.1). (b) Dot plot representation of highlighted fibroblast genes that were significantly upregulated following injury (FDR < 0.1). The color and size of each dot represents the average

*Figure 3 continued on next page*

*Figure 3 continued*

expression and percent of cells expressing each gene, respectively. (**c–d**) Representative images of meningeal whole mounts stained for collagen (green) (**c**) and a 16 color heatmap of the collagen staining intensity (**d**), where red is most intense and blue is least intense. (**e**) Quantification of collagen staining intensity using corrected total cellular fluorescence (CTCF). CTCF is calculated as mean fluorescence of meningeal whole mounts - (Area of meningeal whole mount x Mean fluorescence of background). Each data point represents an individual mouse (Sham n=5, TBI n=5, rep = 1). (**f**) Network map depicting significantly upregulated genes that enriched immune system-related GO terms (FDR < 0.1). The lines within the circle indicate which genes contribute to each GO term. (**g**) Scatter plot representation of the top enriched KEGG terms with significantly upregulated genes in the fibroblast population (FDR < 0.1). Dot size is proportional to term size. Genes contributing to one KEGG term may also contribute to other KEGG terms. (**h**) Dot plot depicting dural, arachnoid, and pial fibroblasts markers where the size of the circles represents the percent of cells expressing each gene. (**i**) Feature plots of genes characteristic of dural fibroblasts in both Sham and TBI conditions. The color of each data point represents the expression level of the indicated gene within that cell. Graphs were calculated using Seurat by normalizing the dataset, finding the variable features of the dataset, scaling the data, and reducing the dimensionality. Differential gene expression was calculated using the ZINB-WaVE function for zero-enriched datasets and DESeq2. Each data point in a UMAP plot represents a cell. Error bars depict mean ±s.e.m. p value for (**e**) was calculated using an unpaired two-tailed t-test assuming unequal variances. FDR; false discovery rate, p.adj; adjusted p-value.

The online version of this article includes the following source data and figure supplement(s) for figure 3:

**Source data 1.** Number of up- and downregulated fibroblast genes one week post-TBI.

**Source data 2.** Raw data for the differential expression analysis in the fibroblast population 1 week post-TBI.

**Source data 3.** Table depicting the corrected total cellular fluorescence for collagen in meningeal whole mounts.

**Source data 4.** Raw data showing enriched KEGG disease processes and contributory differentially expressed fibroblast genes.

**Figure supplement 1.** Ligand-target interactions between fibroblasts and other meningeal cells.

**Figure supplement 1—source data 1.** List of top fibroblast ligands likely to influence the gene expression patterns of the other major cell populations in the scRNA-seq dataset.

after TBI (***Figure 3c, d and e***). Furthermore, we were also interested in determining whether the fibroblast population was contributing to the inflammatory response following TBI. Of the significantly upregulated genes identified in fibroblasts post-TBI, many of them were related to components of immune system activation and cytokine signaling (***Figure 3f***).

To explore the cellular and disease pathways that are most affected in fibroblasts after mild head trauma, we identified the KEGG terms enriched by the differentially upregulated genes in the fibroblast group after TBI in comparison to the Sham group. Interestingly, disease pathways related to neurodegenerative diseases, including Parkinson's disease, Alzheimer's disease, amyotrophic lateral sclerosis, and prion disease, were some of the most highly upregulated pathways altered in fibroblasts after TBI (***Figure 3g***). Many of the same terms that contribute to the oxidative phosphorylation KEGG term also contribute to the various disease-related KEGG terms, indicating a change in the metabolic state of the fibroblasts may be underlying disease-related processes.

Given that fibroblasts are present in all three meningeal layers (***DeSisto et al., 2020***), we decided to investigate which layers the fibroblasts inhabited, and which layer was likely responsible for the increase in fibroblasts following TBI. To accomplish this, we examined the expression of molecules that are commonly used to identify the distinct layer of the meninges in which the fibroblast population is likely to reside (***DeSisto et al., 2020***; ***Doro et al., 2019***; ***Cooper et al., 2012***; ***Zarbalis et al., 2007***; ***Kalamarides et al., 2011***; ***Siegenthaler et al., 2009***; ***Caglayan et al., 2014***). More specifically, recent studies have shown that dural fibroblasts can be identified using the markers *Alpl* and *Foxc2* (***Doro et al., 2019***; ***Cooper et al., 2012***; ***Zarbalis et al., 2007***), whereas *Ptgds* and *Ald1a2* are unique markers of arachnoid fibroblasts (***Kalamarides et al., 2011***; ***Siegenthaler et al., 2009***) and *Col18a1* is specific to pial fibroblasts (***DeSisto et al., 2020***; ***Caglayan et al., 2014***). As expected, we found that a majority of the fibroblasts in the meninges were from the dura, the thickest layer of the meninges and the layer that is targeted by the method of dissection used in these studies (***Alves de Lima et al., 2020b***; ***Rua and McGavern, 2018***). Fewer pia- or arachnoid-resident fibroblasts were present, as anticipated (***Figure 3h***). When we looked at the expression level of dural fibroblast genes before and after TBI, we saw that several of the markers (e.g. *Foxc2*, *Fxyd5*) were significantly upregulated following injury, and other dural markers, while not expressed at significantly higher levels, were clearly expressed by a higher proportion of total cells (e.g. *Alpl*, *Mgp*) (***Figure 3i***). This indicates that the dural compartment likely undergoes an increase in fibroblast cell frequency one week after brain injury, which is also consistent with the increase in collagen density seen in the meninges 1 week after TBI (***Figure 3c, d and e***). Overall, we observed that the meningeal fibroblast population is highly

responsive to TBI and that they upregulate genes enriched in disease-related pathways, immune system activation, and the wound healing response.

## Transcriptional modulation of meningeal lymphocytes in response to mild TBI

Since we observed shifts in the frequencies of some immune cell populations after TBI (*Figure 1d and e*), we were interested in determining which genes were differentially expressed in meningeal T and B cells after head injury, especially given recent reports that have identified instrumental roles for meningeal lymphocytes in regulating multiple aspects of neurobiology, behavior, and CNS disease (*Alves de Lima et al., 2020a*; *Filiano et al., 2016*; *Derecki et al., 2010*; *Ribeiro et al., 2019*; *Gate et al., 2020*). We independently combined the two T cell populations ('Activated T Cells' and 'CD3 + T Cells') and the B cell populations ('B Cells 1', 'B Cells 2', and 'Immature/Differentiating B Cells') to assess differential gene expression. Overall, 151 genes were upregulated and 286 were downregulated following injury in the T cell population, 102 genes were upregulated and 158 were downregulated following injury in the B cell population, and 183 genes were upregulated and 149 were downregulated following injury in the dendritic cell population (*Figure 4a*). Some of the smaller populations such as NK cells, neutrophils, and plasmacytoid dendritic cells exhibited few to no differentially regulated genes, likely due to the small number of cells present in each of these populations (*Figure 4a*).

We were next interested in determining which different T and B cell subsets were present within the meninges, so we re-clustered the cells within these two populations (*Figure 4b and c*). We found that within the T cell subsets, there was a clear CD8 +T cell population and two T helper cell populations: Th2 cells and Th17 cells (*Figure 4b*). The Th2 cell sub-cluster expressed highly-significant cluster-defining markers including *Il1rl1* and *Gata3,* which are characteristic of the Th2 subset (*Tibbitt et al., 2019*; *Figure 4b*, *Figure 4—figure supplement 1a*). Alternatively, the Th17 sub-cluster expressed characteristic markers such as *Il23r*, *Il17re*, and *Rorc* (*Hu et al., 2017*; *Figure 4b*, *Figure 4—figure supplement 1b*). The fourth sub-cluster of T cells appears to be comprised of NKT and NK cells, as this population expressed high levels of common NK markers, including *Klrb1c, Ncr1, Klrd1*, and *Klrk1,* and some of these same cells also expressed components of the CD3 co-receptor (*Cd3d, Cd3d*, and *Cd3g*) (*Figure 4b*, *Figure 4—figure supplement 1c*). The final population represents cells that are likely dying T cells, based on their high expression of mitochondrial genes and *Malat1* (*Figure 4b*).

Re-clustering of the B cell populations revealed five sub-clusters (*Figure 4c*). One sub-cluster appeared to be comprised of mature B cells given its high expression of B cell maturity marker *Cd37* and the B cell receptor components (*Cd79a* and *Cd79b*) (*Figure 4c*; *de Winde et al., 2016*). A second cluster, deemed 'Activated B Cells', was characterized by significant expression of HLA-related genes including *H2-Aa, H2-Eb1,* and *H2-Ab1,* and survival-related genes including *Gimap3, Gimap4*, and *Gimap6*. These activated B cells also highly expressed genes important for adhesion, including *Sell*, which encodes for L-selectin and is a marker for mature B cells (*Lee et al., 2020*; *Figure 4c*, *Figure 4—figure supplement 2a*). A third cluster appeared to be differentiating or immature B cells based on their high expression of *Rag1* and *Rag2* (*Figure 4c*, *Figure 4—figure supplement 2b*). A fourth cluster, deemed 'Proliferating Cells' expressed high levels of *Myc* and *Ccnd2* amongst other cell cycle related genes (*Figure 4c*, *Figure 4—figure supplement 2c*). The final population represents cells that are likely dying B cells due to their high expression of *Malat1* (*Figure 4c*).

In order to determine T and B cell maturation trajectory within the meninges, we performed pseudotemporal analyses using Monocle3 (*Trapnell et al., 2014*). The T cell populations did not demonstrate a strong trajectory in their differentiation status, which is expected given that the populations we identified (Th2, Th17, CD8 + T cells) are all relatively advanced within T cell maturation (*Figure 4d*). However, when we examined the pseudotemporal trajectory of the B cells, we observed a path that confirmed our initial cluster assignments (*Figure 4e*). We observed that the B cells earliest in the differentiation trajectory, as demonstrated by the lowest values on the pseudotime scale, were the 'Immature B Cells' and 'Proliferating Cells' populations, whereas the 'Activated B Cells', that are likely producing antibodies, and 'Mature B cells' were the furthest along in the differentiation trajectory (*Figure 4e*).

Next, we were interested in looking more closely at some of the genes that were significantly upregulated in both the T and B cell populations to determine how these adaptive immune populations

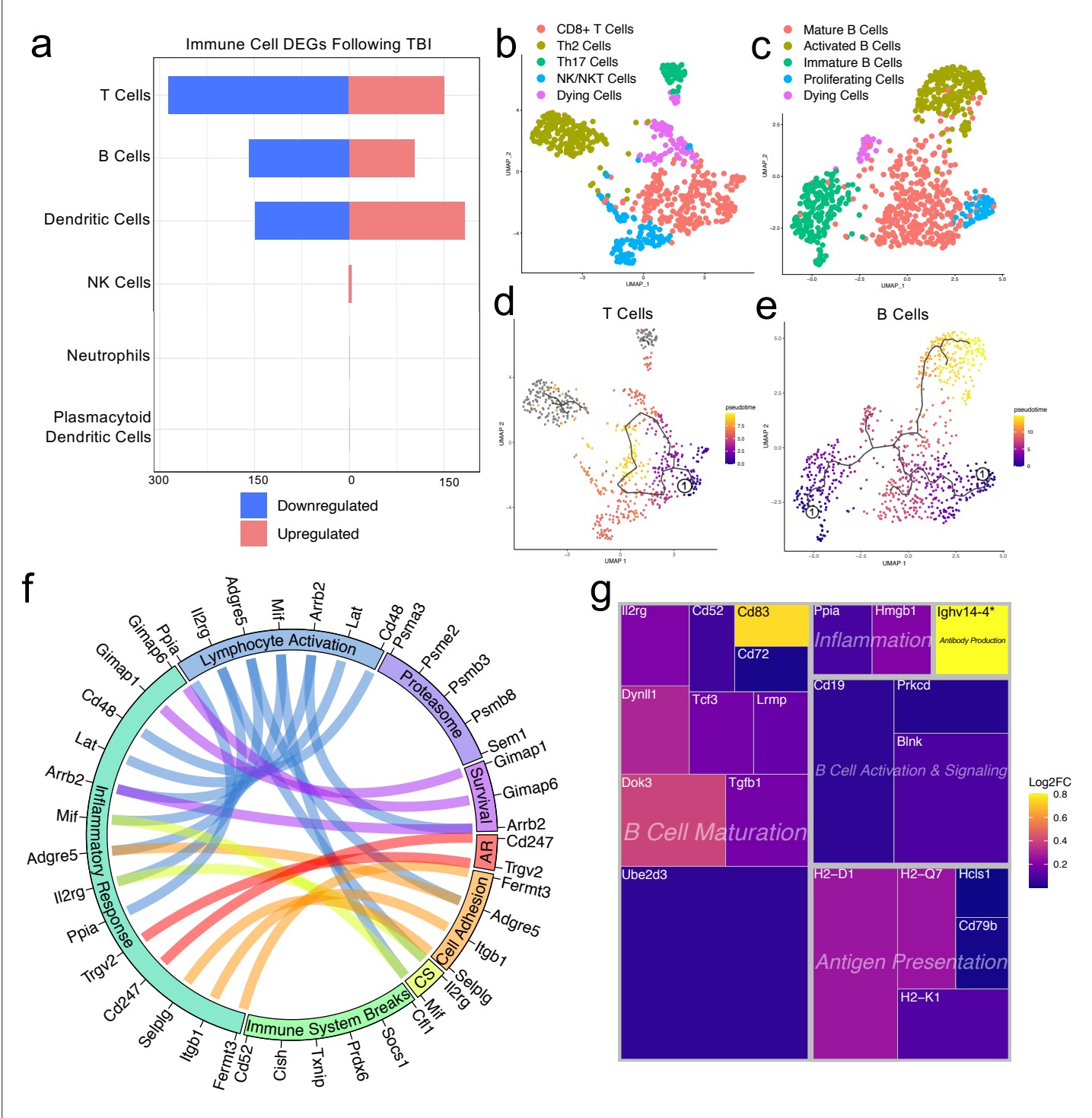

**Figure 4.** Transcriptional response of meningeal lymphocytes to mild TBI. Male WT mice at 10 weeks of age received a TBI or Sham procedure. One week later, the meninges from 5 mice per group were harvested, pooled, and processed for scRNA-seq. (**a**) Quantification of the number of upregulated and downregulated genes in different immune cell populations following injury (FDR <0.1). (**b–c**) UMAP representation showing re-clustering of the (**b**) T cell and (**c**) B cell populations present within the meninges. (**d-e**) UMAP representation of pseudotime cellular trajectory profiles showing (**d**) T cell and (**e**) B cell maturation trajectories. The circle with the number '1' represents the root node. The color of each data point represents advancement in pseudotime, with dark purple representing 'early' pseudotime and yellow representing 'late' pseudotime. The line represents the 'path' of pseudotime with intersections representing possible different differentiation events. Grey data points represent cell populations that were not connected in pseudotime with the selected node. (**f**) Circos plot depicting differentially expressed genes in the T cell populations within the TBI meninges (FDR <0.1)

*Figure 4 continued on next page*

*Figure 4 continued*

associated with different cellular processes. The proportion of the circle's circumference allocated to each cellular process represents the number of T cell genes associated with that process that are differentially expressed in the TBI meninges. The lines connecting genes within the circle indicate which genes were shared amongst cellular processes. Colors were randomly assigned. (**g**) Treemap depicting significantly upregulated genes in the B cell population and the cellular process to which each gene contributes. The size of the square around each gene represents the Wald statistic, which is used to calculate the overall significance of the change in gene expression (a larger square indicates a larger Wald statistic, which leads to a lower adjusted p-value). The color of the boxes represents log2FC, where purple represents a lower log2FC and yellow represents a higher log2FC. An asterisk (*) indicates that the log2FC of the gene was higher than the scale (*Ighv14-4* had a log2FC of 18.08). Graphs were calculated using Seurat by normalizing the dataset, finding the variable features of the dataset, scaling the data, and reducing the dimensionality. Each data point in a UMAP plot represents a cell. Differential gene expression was calculated using the ZINB-WaVE function for zero-enriched datasets and DESeq2. Pseudotime graphs were created using Monocle. AR, antigen recognition; CS, cytokine signaling; FDR; false discovery rate, log2FC; log 2 fold change.

The online version of this article includes the following source data and figure supplement(s) for figure 4:

**Source data 1.** Number of up- and downregulated immune-cell genes 1 week post-TBI.

**Source data 2.** Cluster-defining genes for T cell populations.

**Source data 3.** Cluster-defining genes for B cell populations.

**Source data 4.** Differentially expressed T cell genes contributing to immune-related cellular functions.

**Source data 5.** Raw data for the differential expression analysis in B cells 1 week post-TBI.

**Figure supplement 1.** Cluster-defining genes for T cell subpopulations.

**Figure supplement 2.** Cluster-defining genes for B cell subpopulations.

were affected following injury. The T cell populations upregulated many genes important for survival (*Gimap1, Gimap6*), activation (*Arrb2, Ppia, Cd48*), cytokine signaling (*Mif, Il2rg*), and antigen recognition (*Cd247*), all of which contributed to an overall increase in inflammatory response gene expression (*Figure 4f*). Concomitantly, the T cells also upregulated various genes that are known to be involved in the dampening of immune responses such as *Socs1* (Suppressor of Cytokine Signaling-1) and *Cd52* (*Figure 4f*; *Liau et al., 2018*; *Toh et al., 2013*).

Investigating the genes that were upregulated in the B cell populations following injury, we found that many of these genes fell into the category of 'B Cell Maturation', including *Cd83, Ube2d3*, and *Doc3* (*Figure 4g*). Other upregulated genes included those important for B cell activation and signaling (*Blnk, Cd19*), antigen presentation (*Cd79b, H2-D1*), and inflammation (*Ppia, Hmgb1*) (*Figure 4g*). The upregulation of these genes suggests that TBI drives the activation and maturation of B cell populations in the meningeal compartment. Overall, these data demonstrate that both T and B cells upregulate genes involved in activation of adaptive immune responses following head trauma. This upregulation seems to be controlled, as multiple regulatory genes are also simultaneously activated.

## Predicted ligand-target interactions highlight a pro-growth and controlled pro-inflammatory meningeal environment after TBI

Cell-cell interactions may partly dictate the gene expression dynamics observed following TBI. We used Nichenet to explore how intercellular communication may influence the post-injury transcriptional environment. Nichenet is a tool that integrates gene expression data from interacting cells to infer the effects of sender-cell ligands on receiver cell expression (*Browaeys et al., 2020*). Since both macrophage and fibroblast populations exhibited significant transcriptional alterations post-TBI we focused our analysis on how signaling of these large populations affects other meningeal cells (*Figure 1e*).

We first examined how the macrophage cell populations might influence the gene expression patterns of the other major cell populations in the scRNA-seq dataset (T cells, B cells, Dendritic Cells, NK cells, Fibroblasts, and Endothelial Cells) through inferred ligand-target interactions. Of all macrophage ligands, *Tgfb1* best predicted the gene expression patterns seen in other cell populations, suggesting the observed transcriptional dynamics reflect an environment of cell growth, differentiation, and alternative macrophage activation (*Figure 2—figure supplement 2a and b*; *Gong et al., 2012*). Furthermore, other top-predicted macrophage ligands including *Itgam, Apoe, Vcam1, Selplg, Nectin1*, and *Itgb1* play critical roles in cell adhesion and phagocytosis, both necessary for mounting an inflammatory response (*Figure 2—figure supplement 2a and b*; *Grainger et al., 2004*). Other top-ligands with pro-inflammatory properties include *Adam17*, which is involved in the processing of

TNF at the surface of the cell, *Tnfsf13b*, which promotes activation and proliferation of B cells, and *C3*, which is part of the complement cascade (*Black et al., 1997*). We observe gene expression changes and cellular phenotypes consistent with predicted ligand effects on target cells. For example, *Tgfb1* signaling is predicted to activate multiple genes that affect collagen production including *Col1a1*, *Col1a2* and *Col3a1* (*Figure 2—figure supplement 2c*), aligning with our finding of increased collagen production following TBI (*Figure 3c–e*). Predicted target genes of other pro-inflammatory ligands such as *Adam17* and *Tnfsf13b* include B cell response genes (*Cd19, Cd79a,* and *Ighm*) and genes related to immune cell activation and antigen-presentation (*Cd38* and *Cd40*) (*Figure 2—figure supplement 2c*). Finally, when we examined how ligands from other cells in the meninges may impact the macrophage gene signature, we saw that ligands from multiple cell populations (Endothelial Cells, Dendritic Cells, T cells, and B cells) are likely responsible for shaping this signature (*Figure 2—figure supplement 2d*). This is not surprising given that monocytes and macrophages likely interact with all of the meningeal cell populations and play very important roles in shaping the overall gene expression signatures seen after TBI. Overall, the predicted ligand-target interactions between the macrophage population and other major meningeal cell populations after brain injury illustrate a proliferative, pro-inflammatory state.

Next, we examined how the meningeal fibroblast population might influence local gene expression through ligand-target interactions. As with the macrophage ligands, we found the top ligand signaling pathways in fibroblasts were essential for cellular growth and differentiation. More specifically, many top ligands were essential for promoting angiogenesis (*Vegfa, Cxcl12, Pgf*), growth and inflammation (*Apoe, Csf1, Cxcl12, Tgfb3*), and extracellular matrix development and wound healing (*Col4a1, Hspg2, Nov*) (*Figure 3—figure supplement 1a and b*; *Solé-Boldo et al., 2020*; *Buechler et al., 2021*; *Lin et al., 2005*). Interestingly, we found anti-inflammatory pathways among the top predicted ligands including *Anxa1*, which is known for its inhibitory effects on adhesion and migration, and *Serping1*, which is responsible for production of C1 inhibitor (*Figure 3—figure supplement 1a and b*; *Gavins and Hickey, 2012*). Predicted targets of the top ligand, *Apoe*, include complement cascade genes such as *C1qb* and *C1qc*, both of which were upregulated in the meningeal macrophage population after injury. Other ligands, such as *Csf1*, also likely potentiate the immune response by activating genes such as *Cd14* and *Cd68*, which are important for macrophage/monocyte responses in inflammation (*Figure 3—figure supplement 1c*; *Buechler et al., 2021*). Other ligands, such as *Vegfa*, likely activate endothelial cells, which results in upregulation of genes important for cellular growth and division including *Fos* and *Id1* (*Figure 3—figure supplement 1c*). Upon performing the inverse analysis, we find that endothelial cells and macrophages strongly influence the fibroblast transcriptome as evidenced by high expression of influential ligands in a significant proportion of the cell population (*Figure 3—figure supplement 1d*). Altogether, meningeal fibroblasts likely influence gene expression signatures after TBI to promote pro-growth and angiogenic signaling cascades in addition to a balanced upregulation of the immune system. Together with macrophages, these predicted ligand-target interactions highlight the highly complex but coordinated and controlled response to injury that occurs in young mice.

## Prominent effects of aging and mild TBI on the meningeal transcriptome

Given the considerable brain injury-induced alterations in the meningeal transcriptional and cellular landscape that we observed in young mice, we were next interested in investigating whether these changes were preserved or altered with age. It has previously been suggested that inability to properly resolve inflammatory responses in the brain after head trauma contributes to the aggravated disease course commonly seen in the elderly (*Chou et al., 2018*; *Witcher et al., 2021*; *Witcher et al., 2018*). Therefore, we were also particularly interested to explore potential differences in the resolution of meningeal immune responses following TBI between young and aged mice. To this end, we performed bulk RNA-seq on the meningeal tissue at 1.5 months post TBI or Sham treatment in both young (10 weeks of age) and aged (20 months of age) mice, as we predicted that the meningeal injury would have largely resolved 1.5 months post-TBI in young mice (*Figure 5a*). Principal component analysis (PCA) revealed that age was the main driver of differential gene expression, as young and aged groups clustered furthest apart (*Figure 5b*). However, while the young mice that had received either Sham or TBI clustered together in the PCA plot, the aged Sham and TBI mice clustered further apart,

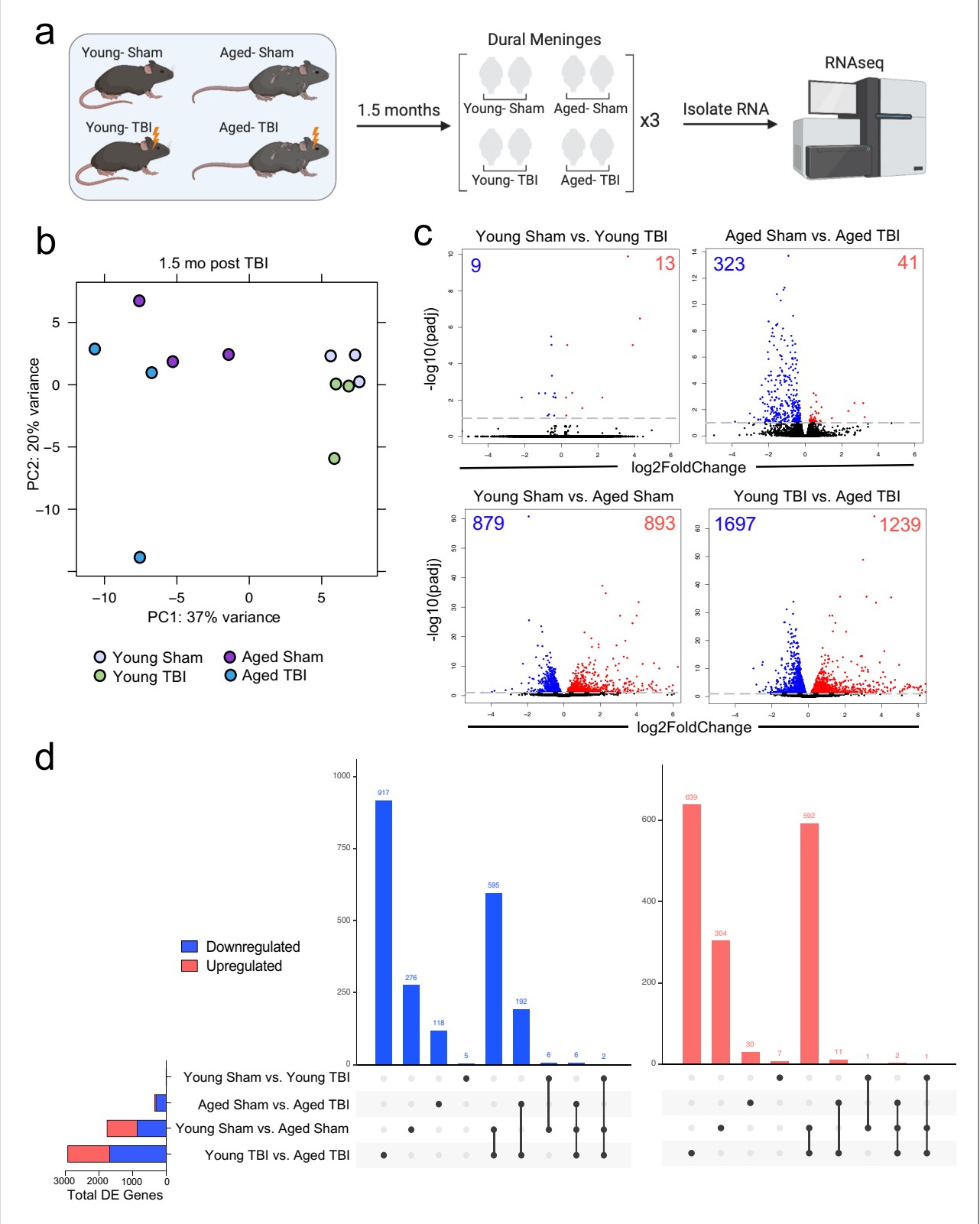

**Figure 5.** Effects of aging and mild TBI on the meningeal transcriptome. (**a**) Schematic depicting experimental layout. Male WT mice at 10 weeks of age or 20 months of age received a TBI or Sham procedure. 1.5 months later, bulk RNA-seq was performed on the four experimental groups with three biological replicates per group (each biological replicate consisted of meningeal RNA samples from 2 to 3 independent mice). (**b**) Principal component analysis (PCA) showing clustering of samples. (**c**) Volcano plots illustrate the number of differentially expressed genes with statistically significant

*Figure 5 continued on next page*

*Figure 5 continued*

differences denoted in blue and red (FDR <0.1). Numbers in each corner depict the number of differentially expressed genes for each comparison. Blue data points represent significantly downregulated genes and red data points represent significantly upregulated genes. (**d**) Upset plots depicting significantly downregulated (blue) and upregulated (red) genes for each comparison and the number of genes that were shared between comparisons (FDR <0.1). The graph on the left sidebar shows the total number of differentially expressed genes per group. A single black dot indicates the differentially expressed genes are unique to the highlighted comparison. Two or more black dots connected by a line indicate that the differentially expressed genes are shared between the multiple highlighted comparisons. FDR and p-values were calculated with DESeq2 using the Wald test for significance following fitting to a negative binomial linear model and the Benjamini-Hochberg procedure to control for false discoveries. FDR; false discovery rate, DE; differentially expressed.

The online version of this article includes the following source data for figure 5:

**Source data 1.** Significantly up- and downregulated genes for each bulk RNA-seq comparison.

**Source data 2.** Raw data of shared and unique differentially expressed genes for each comparison.

indicating that the effects from TBI may have more long-lasting effects on gene expression in aged mice when compared to young mice (*Figure 5b*). Indeed, when we looked at the number of differentially expressed genes between all four experimental groups, we saw that there were only a total of 22 differentially expressed genes when comparing Young Sham with Young TBI, while there were a total of 364 differentially expressed genes when comparing Aged Sham with Aged TBI (*Figure 5c and d*). Interestingly, 1772 differentially expressed genes were identified when comparing Young Sham mice with Aged Sham mice, and 2936 differentially regulated genes were identified when comparing Young TBI mice with Aged TBI mice (*Figure 5c and d*). Furthermore, we looked at which genes were shared between comparison groups to determine if the TBI signature in aging was unique or largely shared with the uninjured aged mice. Interestingly, there were 917 downregulated genes and 639 upregulated genes that were unique to the Young TBI vs. Aged TBI comparison and not shared with any other comparison, including the Young Sham vs. Aged Sham comparison. There were 595 downregulated genes and 592 upregulated genes that were shared between these two comparisons, indicating that while a portion of the transcription changes seen in the Young TBI vs. Aged TBI group may be attributed to aging, a significant number of the affected genes were uniquely identified in the setting of trauma in aged mice. Overall, this indicates that aging profoundly affects meningeal gene expression and that mild head trauma in aging results in even larger changes in gene expression. Moreover, while the young mice exhibit very few gene expression changes 1.5 months following TBI, the aged mice experience many more alterations in gene expression that last for a longer period of time, which suggests that recovery post-TBI may be delayed with aging.

Because aging itself resulted in substantially different gene expression patterns, we decided to look more closely at these differences within the bulk RNA-seq dataset. Upon examining the top 20 most significantly upregulated and downregulated genes in the Aged Sham mice as compared to the Young Sham mice, we noticed a striking upregulation in genes important for antibody production by B cells (*Figure 6a*). In fact, one half of the top 20 upregulated genes fell into this category (*Figure 6a*). When we examined the top GO biological processes that were enriched by the significantly activated genes in the Young Sham mice versus Aged Sham mice comparison, we saw that immune and defense responses were among the most highly upregulated (*Figure 6b*), indicating that the cells within the aged meninges have grossly upregulated their immune response, even in homeostatic conditions.

Due to the striking nature of the upregulation of antibody production-related genes, and recent findings that report an increase in IgA-secreting plasma cells with age (*Fitzpatrick et al., 2020*), we more closely examined some of these genes within the bulk RNA-seq dataset (*Figure 6c*). We found highly significant upregulations in genes related to the immunoglobulin heavy chain (*Ighm, Ighg2b, Igha*), light chain (*Igkc*), and components of IgA or IgM antibodies (*Jchain*) (*Figure 6c*). Using immunohistochemistry on meningeal whole mounts, we confirmed that aged meninges have significantly increased J chain deposition that is concentrated along the sinuses (*Figure 6d and e*). Next, we wanted to determine whether the increased antibody-related gene production we saw in aged mice was due to an overall increased number of B cells. Interestingly, we saw that the number cells expressing B220 along the meningeal transverse sinus in mice was significantly lower in aged mice (*Figure 6f and g*), which is in contrast to other recent studies have shown that B cells comprise a larger proportion and number of cells in aged dural meninges (*Brioschi et al., 2021*; *Mrdjen et al., 2018*). Therefore, as a second method of validation, we performed flow cytometry on the entire

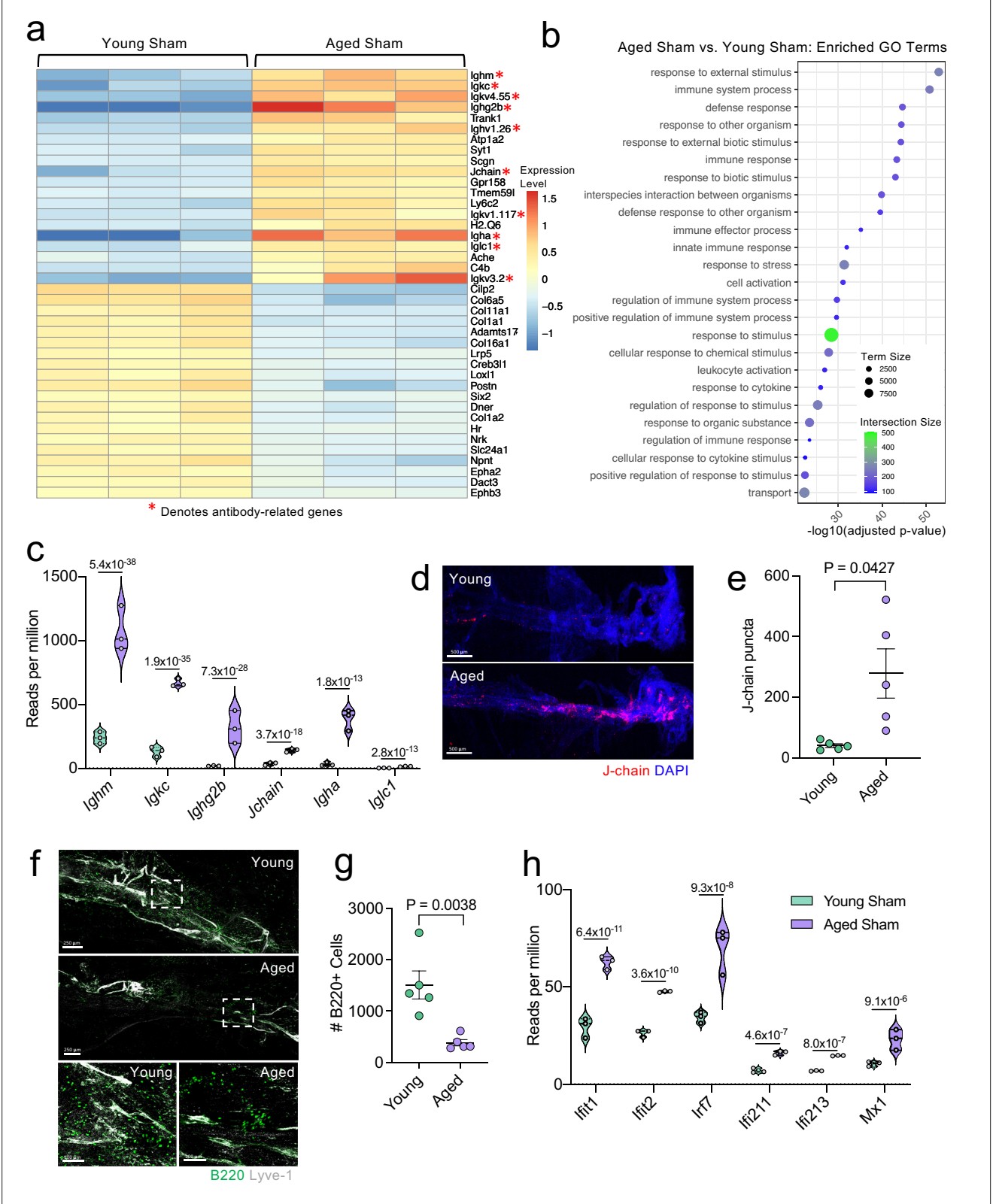

**Figure 6.** Aging promotes the upregulation of meningeal genes involved in type I IFN and antibody signaling. Male WT mice at 10 weeks of age or 20 months of age received a TBI or Sham procedure. 1.5 months later, bulk RNA-seq was performed on the four experimental groups with three biological replicates per group (each biological replicate consisted of meningeal RNA samples from 2 to 3 independent mice). (**a**) Heatmap representation of the top 20 most significantly upregulated and downregulated (FDR <0.1) genes in the Young Sham vs. Aged Sham groups. The red

*Figure 6 continued on next page*

*Figure 6 continued*

asterisk (*) indicates genes associated with antibody production. (**b**) Dot plot of GO term biological processes shows enrichment of immune-related pathways with differentially expressed genes between young mice as compared to aged mice. Color and size of each dot represent the size of the GO term and the number of upregulated genes that contribute to each term, respectively. (**c**) Violin plot depicting counts of significantly activated antibody and B cell related genes in response to age (FDR <0.1). The number above each comparison on the graph represents the adjusted p-value calculated for each gene using DESeq2. The central line within each plot represents the median of the data set. The upper and lower boundaries of the box represent the third (Q3) and first (Q1) quartiles respectively. The violin plot encompasses the three biological replicates. The width of the violin plot represents the frequency of observations at that given y-value. Therefore, the wider the violin plot, the higher the frequency of observations. The meninges of 5 young Sham mice and 5 aged Sham mice were harvested for each immunohistochemical experiment. (**d**) Representative images from a young Sham mouse and aged Sham mouse showing a region of the SSS stained with J-chain (red) and Lyve-1 (grey) (**e**) and quantification of J-chain puncta in meningeal whole mounts along the SSS (Sham n=5, TBI n=5, rep = 1). (**f**) Representative images of the transverse sinus in young and aged mice stained with B220 (green) and Lyve-1 (grey). The dashed box on the top two images corresponds to the higher magnification images depicted below. (**g**) Quantification of the number of B220 cells along the entire transverse sinus (Sham n=5, TBI n=5, rep = 1). (**h**) Violin plot depicting counts of significantly activated type-I interferon related genes in response to age (FDR < 0.1). The violin plot parameters are the same as describe for (**c**). FDR and p-values in (**a–c,h**) were calculated with DESeq2 using the Wald test for significance following fitting to a negative binomial linear model and the Benjamini-Hochberg procedure to control for false discoveries. Error bars in (**e,g**) depict mean ±s.e.m. p values in (**e,g**) were calculated using a two-tailed unpaired two-sample t-test assuming unequal variances. FDR; false discovery rate, SSS; superior sagittal sinus.

The online version of this article includes the following source data and figure supplement(s) for figure 6:

**Source data 1.** Table depicting all differentially regulated genes in the Young Sham vs. Aged Sham comparison.

**Source data 2.** Raw data depicting enriched GO-terms and contributory genes that were upregulated in aging.

**Source data 3.** Number of reads per million of antibody-related genes upregulated in aging in the bulk RNA-seq dataset.

**Source data 4.** Table depicting the number of J-chain puncta quantified in young and aged mice.

**Source data 5.** Table depicting the number of B220 + cells along the transverse sinus in young and aged mice.

**Source data 6.** Number of reads per million of interferon-related genes upregulated in aging in the bulk RNA-seq dataset.

**Figure supplement 1.** Aging does not appreciably influence the total numbers of meningeal CD19 + and B220 + cells.

**Figure supplement 1—source data 1.** Tables depicting flow cytometry cell counts and frequencies of B cells in the dural meninges in young and aged mice.

dural meninges to assess for B220 + and CD19 + cell counts and frequency within total CD45 + cells (*Figure 6—figure supplement 1a*). The aged dural meninges had overall fewer cells when compared to young dural meninges (*Figure 6—figure supplement 1b*). Upon examination of total B220+, CD19+, and B220 + CD19 + cell numbers, we did not see significant differences in aged mice (*Figure 6—figure supplement 1c*). While not significant, the overall frequency of B220+, CD19+, and B220 + CD19 + cells was decreased in aged dural meninges when compared to young counterparts (*Figure 6—figure supplement 1d*). In summation, we do not see increases in B cell numbers in the aged dura, which suggests that the increase in antibody-related gene expression seen in aging may reflect a change in the function of the B cells in the dural meninges rather than a recruitment of more B cells. Differences in our data compared to other published findings may reflect regional differences in B cell populations along the transverse sinus given the impaired meningeal lymphatics seen in aged mice. Other potential drivers of our distinct findings include differences in sex and microbiome, both of which might also influence the number and frequency of B cells in the meninges. Overall, this suggests that the composition of the B cell population in aged mice may be substantially different than in young mice; however, future studies are needed to formally evaluate this in greater detail.

In addition to the antibody-related gene upregulation, we also observed increased expression of type I interferon (IFN)-related genes within the bulk RNA sequencing dataset (*Figure 6h*). Type I IFNs have been shown to be upregulated in the brain parenchyma in various neurological disorders, where they are generally thought to play deleterious roles in promoting disease pathogenesis (*Karve et al., 2016*; *Barrett et al., 2020*; *Baruch et al., 2014*; *Abdullah et al., 2018*; *Zhang et al., 2017*). Our data indicate that this type I IFN signature is also seen within the meningeal compartment of aged mice. Amongst others, we saw highly significant increases in type I IFN related genes including *Ifit1*, *Ifit2*, *Irf7*, *Ifi213*, and *Mx1* (*Figure 6h*). These findings demonstrate that aging promotes profound alterations in the meningeal transcriptome. Moreover, the upregulation of antibody genes and type I IFN related-genes suggests an overall elevation in immune activation status in the aged meninges.

## Injury in aged mice results in prolonged inflammatory responses

In order to assess the unique transcriptional response to TBI in aged compared to young mice, we analyzed the transcriptional response that is exclusive to the Young TBI vs Aged TBI comparison and not shared with the Young Sham vs Aged Sham comparison in the bulk RNA-seq dataset. This includes the 917 downregulated genes and 639 upregulated genes that were unique to the Young TBI vs. Aged TBI comparison (*Figure 5d*). While aging and TBI each individually lead to changes in gene expression which have some overlap, the double hit of TBI with old age was found to induce an even larger change in gene expression than either condition alone.

Using GO molecular function terms, we saw that of the repressed genes unique to the Young TBI vs. Aged TBI comparison, many of these genes are involved in binding processes, including protein binding and cytoskeletal binding (*Figure 7a*). When we looked more closely at the top repressed genes unique to the Aged TBI versus Young TBI comparison, we observed that many of these genes encode for collagenases (*Col4a1, Col4a2,* and *Col5a2*) and other molecules involved in regulating cellular junctions (*Jup*) (*Figure 7b*). Using qPCR, we validated that some of these genes important for collagenase production were indeed downregulated in aged mice 1.5 months after TBI in the dural meninges (*Figure 7c*). These pathways likely aid in the wound healing response of the meninges and were upregulated in the response to TBI in young mice (*Figure 3*), however are downregulated after brain injury in aging.

Additionally, we looked into the genes that were uniquely activated in the Aged TBI mice as compared to the Young TBI mice. We found that genes associated with immune activation were profoundly upregulated in aged TBI mice in comparison to their young TBI counterparts (*Figure 7d*). The most enriched GO biological processes included the 'defense response' and 'immune system process' (*Figure 7d*). Some of the genes that contributed to the upregulation of these immune-related terms included those associated with immunoglobulin production (*Ighg2c*), T and B cell signaling (*Cd24a, Zap70, Cxcr6*), and cell death (*Casp12, C2*) (*Figure 7e*). In summary, these findings highlight some of the distinct changes seen in the aged meningeal tissue following TBI. Specifically, we find that mild TBI in aged mice results in prolonged activation of immune genes and decreased expression of genes involved in extracellular matrix remodeling and the maintenance of cellular junctions. Further-more, we report that while the meningeal transcriptome in young mice returns almost completely to baseline resting levels by 1.5 months post mild head injury, the aged meninges, in contrast, continue to exhibit substantial and protracted transcriptional alterations related to head injury.

## Immune system-related transcriptional changes link TBI-driven gene expression patterns in young mice with chronic changes seen in aging

Finally, we wanted to investigate whether post-TBI gene expression changes persist in aged mice by identifying common gene expression patterns between the young mice one week after injury and the aged mice 1.5 months after injury. To do this, we looked at both the scRNA-seq dataset (one week after TBI) and the bulk RNA-seq dataset (1.5 months after TBI) to determine whether there were shared differentially expressed genes. First, we compared the differentially expressed genes in the T cells, B cells, Fibroblasts, and Macrophages in the scRNA-seq dataset with the differentially expressed genes in the Young Sham vs. Aged Sham comparison, to determine whether some of these genes were shared with aging alone (*Figure 8a*). While there were 139 shared differentially regulated genes, a majority of the differentially expressed genes in each dataset were not shared (*Figure 8a*). When we looked more closely at the shared downregulated genes, we found that many were important for wound healing and maintenance of connective tissue (*Figure 8c*). These data suggest that some of the downregulated genes important for wound healing in the subacute time point after TBI remain chronically downregulated in the aged meninges, further supporting the idea that aged meninges may be less able to respond to injury at baseline. Some of the upregulated genes that were shared at the subacute time point and chronically in aging were genes that contribute to abnormal immune cell physiology, innate immune response, and immune cell activation, again supporting the notion that aged meninges adopt a chronic, baseline activation of the immune system (*Figure 8c*). Alternatively, while the young meninges initially upregulate some genes important for an inflammatory response one week following TBI, gene expression levels eventually return to baseline.

Looking more closely at the differentially regulated genes shared between the Young TBI vs. Aged TBI bulk RNA sequencing comparison and the T cells, B cells, Fibroblasts, and Macrophages of the

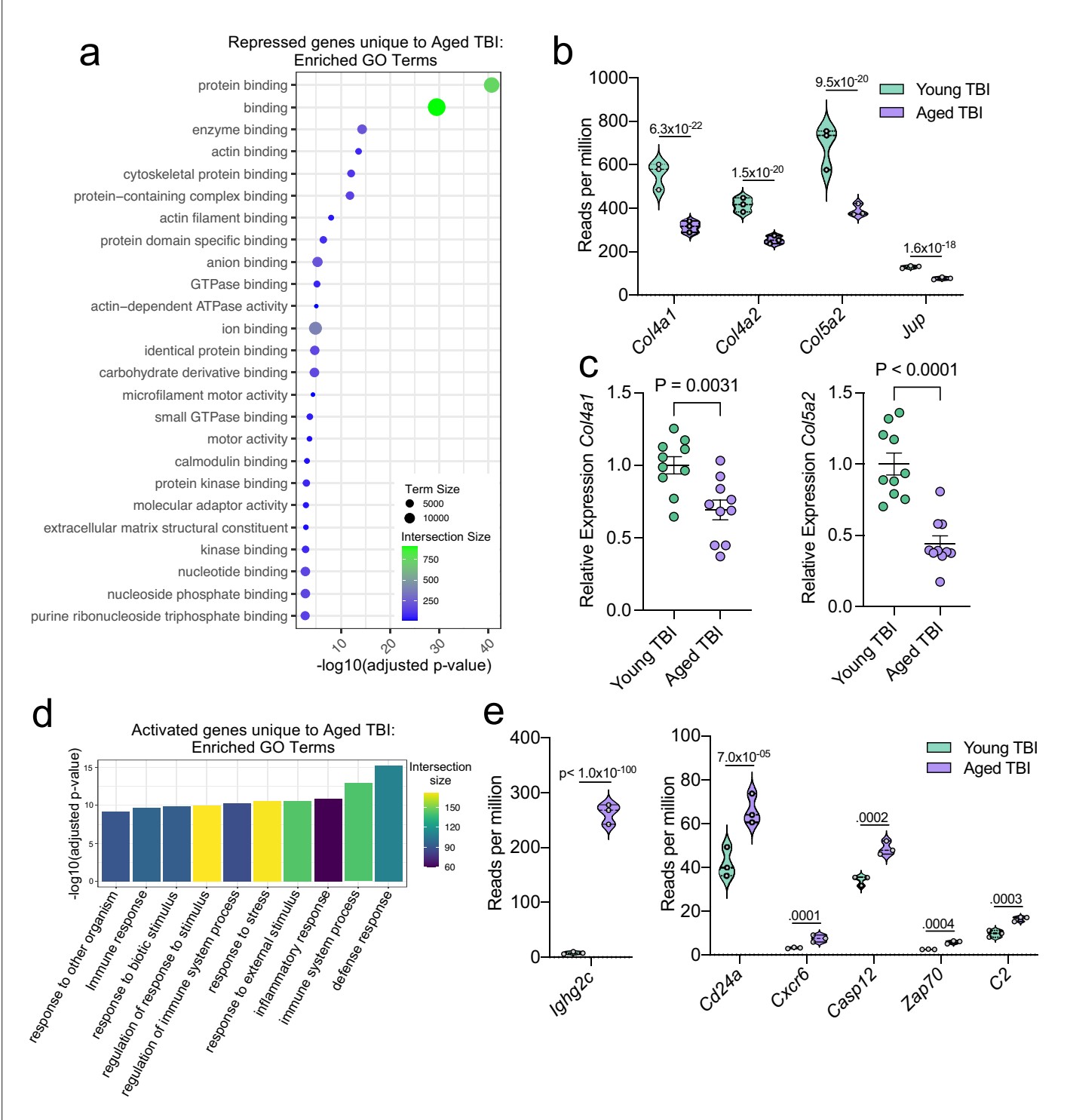

**Figure 7.** Aging and mild TBI together promote a unique meningeal transcriptional signature. Male WT mice at 10 weeks of age or 20 months of age received a TBI or Sham procedure. 1.5 months later, bulk RNA-seq was performed on the 4 experimental groups with 3 biological replicates per group (each biological replicate consisted of meningeal RNA samples from 2 to 3 independent mice). (**a**) Dot plot showing GO term molecular functions enriched by the repressed genes unique to the Young TBI vs Aged TBI comparison. The color and size of each dot represents the size of the GO term and the number of upregulated genes that contribute to each term, respectively. (**b**) Violin plot depicting counts of significantly repressed extracellular matrix related genes (FDR <0.1). (**c**) Quantitative PCR relative expression of *Col4a1* and *Col5a2* within the dural meninges 1.5 months after TBI (Sham n=10, TBI n=10, rep = 1). (**d**) Bar plot shows enrichment of GO term biological processes related to the immune system with the genes unique to the

*Figure 7 continued on next page*

Figure 7 continued

Young TBI vs Aged TBI comparison. The color of each bar represents the number of upregulated genes that contribute to each GO term. (**e**) Violin plots depicting counts of significantly activated immune-related genes (FDR <0.1). (**b,e**) Each statistic represents the adjusted p-value calculated for each gene using DESeq2. The central line within each plot represents the median of the data set. The upper and lower boundaries of the box represent the third (Q3) and first (Q1) quartiles respectively. The violin plot encompasses the three biological repeats. The width of the violin plot represents the frequency of observations at that given y-value. Therefore, the wider the violin plot, the higher the frequency of observations. FDR and p-values for (**a,b,d,e**) were calculated with DESeq2 using the Wald test for significance following fitting to a negative binomial linear model and the Benjamini-Hochberg procedure to control for false discoveries. Error bars in (**c**) depict mean ±s.e.m. p values for (**c**) were calculated using unpaired two-sample students t-tests.

The online version of this article includes the following source data for figure 7:

**Source data 1.** Raw data showing enriched GO-terms and contributory genes that were uniquely downregulated in aging after TBI.

**Source data 2.** Number of reads per million of collagen-related genes downregulated in aging after injury in the bulk RNA-seq dataset.

**Source data 3.** Relative expression of collagen-related genes by qPCR in whole meninges one week post-TBI.

**Source data 4.** Table depicting all differentially regulated genes unique to the Young TBI vs. Aged TBI comparison.

**Source data 5.** Number of reads per million of immune-related genes downregulated in aging after injury in bulk RNA-seq dataset.

scRNA-seq dataset, we found 119 genes in common (*Figure 8b*). Similar to aging alone, many of the common upregulated genes were related to abnormal immune cell activation, reflecting the chronically activated immune response that occurs after TBI in aging (*Figure 8d*). Of the shared downregulated genes, many contribute to cell adhesion and response to endoplasmic reticulum stress (*Figure 8d*). Altogether, while a majority of the genes that were differentially expressed in both the bulk RNA and scRNA sequencing datasets were not shared, the common genes reflect a pattern of abnormal immune cell activation and a defective response to healing as demonstrated by the downregulation of genes important for extracellular matrix repair and cellular adhesion. While the aged mice still exhibit this signature 1.5 months after injury, the young mice express these shared genes initially, but return to baseline levels 1.5 months after injury.

## Discussion

Findings from these studies highlight the heterogeneous and dynamic nature of the meninges in response to TBI and aging. Following TBI in young mice, there is an enrichment of fibroblast and macrophage populations in the dural meninges, as well as an upregulation in genes associated with immune activation. Interestingly, the gene expression patterns of the meninges are significantly altered in aging, with large upregulations in genes involved in immunoglobulin production and type I IFN signaling. More than a month after injury, the aged meninges exhibit downregulation of genes related to collagenase production, extracellular matrix maintenance, and cell junction formation, when compared to young meninges. The aged meninges after injury also show upregulation of genes involved in immune signaling. Moreover, the aged meninges experience a much more prolonged and substantial response to injury than the meninges in young mice, which have largely returned to baseline by 1.5 months post-injury.

While our overall knowledge of meningeal biology in TBI remains limited, recent work has begun to uncover roles for meningeal macrophages in head trauma (*Russo et al., 2018*; *Roth et al., 2014*). For instance, studies by Roth et al. demonstrated that the release of reactive oxygen species (ROS), which occurs as part of meningeal macrophage cell death, can trigger subsequent tissue damage in the brain parenchyma (*Roth et al., 2014*). Interestingly, they showed that blocking ROS release by dying meningeal macrophages is also effective in minimizing cell death in the brain parenchyma. Collectively, these results suggest that the meningeal response to TBI directly affects cells in the brain as well (*Roth et al., 2014*). More recently, this same group also demonstrated that meningeal macrophages play important roles in coordinating meningeal remodeling and vascular repair following mild head trauma (*Russo et al., 2018*). Here, they showed that distinct macrophage populations function within defined regions in and around the injury site. For example, they identified that inflammatory myelomonocytic cells work in the core of the lesion where dead cells are abundant, whereas wound-healing macrophages are present along the perimeter of the injury where they work to restore blood vasculature and clear fibrin (*Russo et al., 2018*).

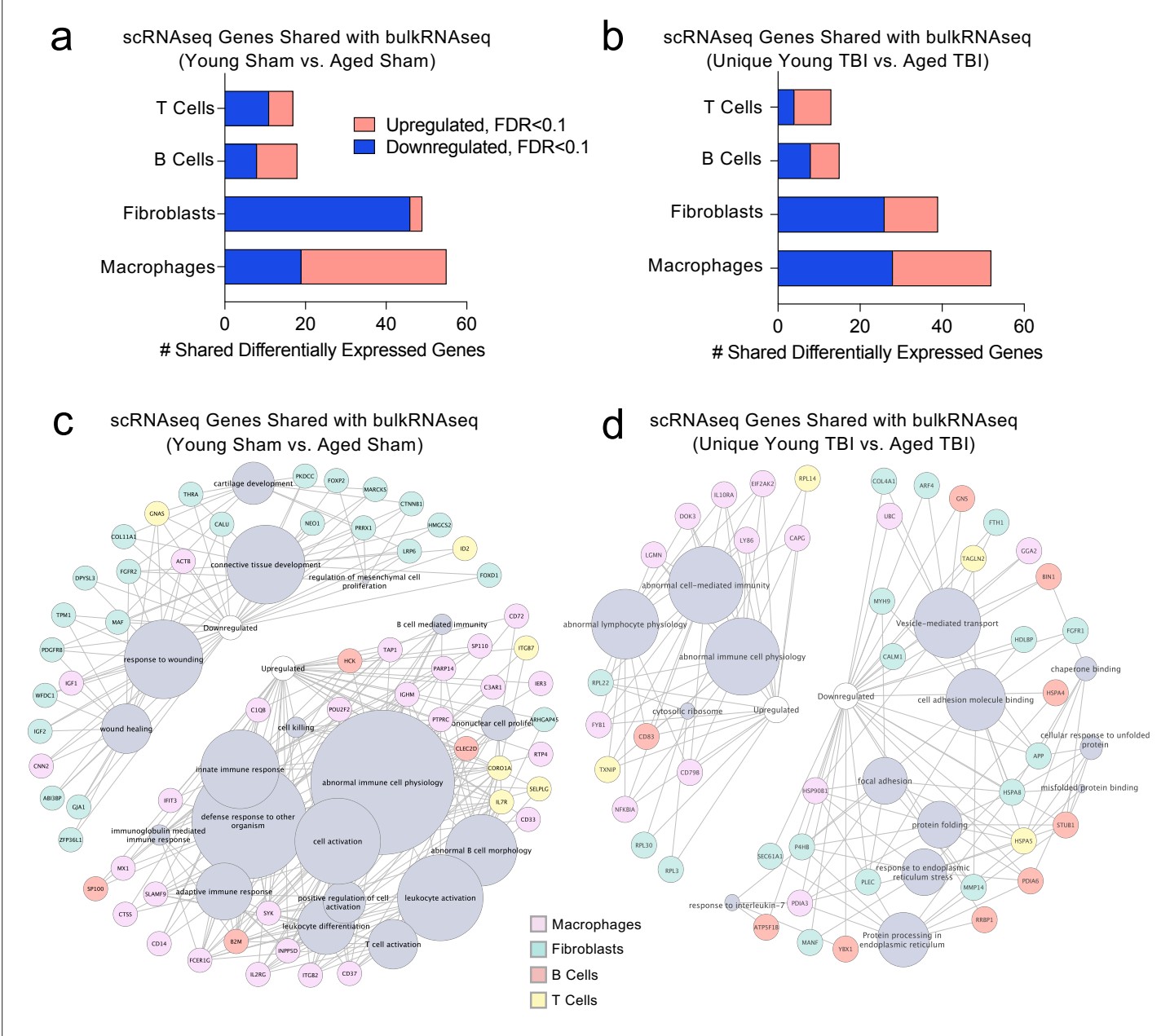

**Figure 8.** Shared gene signatures show dysregulated immune activation. Differential gene expression was compared between the scRNA-seq dataset in young mice (1 week post-TBI) and the bulk RNA-seq dataset in aged mice (1.5 months post-TBI). Quantification of the number of differentially regulated genes shared between T cells, B cells, Fibroblasts, and Macrophages in the scRNA-seq dataset with the differentially expressed genes seen in (**a**) aging alone and (**b**) 1.5 months after TBI in aged mice (FDR < 0.1). Modified circos plots depicting the shared up- and down-regulated genes and the cellular processes to which they contribute, between the T Cells, B Cells, Fibroblasts, and Macrophages from the scRNA-seq dataset and the bulk RNA-seq dataset in (**c**) aging alone and (**d**) 1.5 months after TBI. The color of the circle with each gene represents the cell population to which the gene belongs. The size of the grey circles corresponds to the number of genes contributing to the term, which is shown by the intersecting line from each gene. FDR values in the bulk RNA-seq dataset were calculated with DESeq2 using the Wald test for significance following fitting to a negative binomial linear model and the Benjamini-Hochberg procedure to control for false discoveries. Differential gene expression in the scRNA-seq dataset was calculated using the ZINB-WaVE function for zero-enriched datasets and DESeq2. Graphs in (**c–d**) were constructed using ToppCluster and Cytoscape. FDR; false discovery rate.

The online version of this article includes the following source data for figure 8:

**Source data 1.** Raw data showing differentially expressed genes in scRNA-seq dataset and bulk RNA-seq dataset used for comparisons.

Recent studies have also begun to uncover critical roles for meningeal lymphocytes in various neurological disorders as well as in the regulation of basic neurological functions and behavior (*Alves de Lima et al., 2020b*; *Alves de Lima et al., 2020a*; *Filiano et al., 2016*; *Derecki et al., 2010*; *Ribeiro et al., 2019*; *Rua and McGavern, 2018*; *Brioschi et al., 2021*; *Mrdjen et al., 2018*; *Louveau et al., 2018*). Yet, surprisingly little is currently known with regard to how head trauma impacts adaptive immunity in the meninges. Here, we show that adaptive immune cells found in the meninges of young mice upregulate genes essential for activation and maturation at one week post-TBI. Furthermore, in both aging alone and in aged mice following brain injury, we observed an upregulation of many genes important for antibody production by B cells. Interestingly, emerging studies have proposed that IgA-producing plasma B cells survey the meninges and that their IgA secretion provides an 'immuno-logical barrier' to prevent potential pathogens from gaining access into the brain parenchyma (*Fitz-patrick et al., 2020*). While speculative, perhaps this upregulation in antibody production genes in the meninges is a protective strategy that is mobilized to counteract the loss of blood-brain barrier (BBB) integrity that can occur both in aging and following TBI (*Ren et al., 2013*; *Yang et al., 2020*). Nevertheless, further studies are required to follow up on the significance and function of this eleva-tion in antibody-related genes that is seen in the meninges as a result of aging and head trauma.

For the studies that appear in this paper we paid special attention to the transcriptional response that occurs in meningeal macrophages, fibroblasts, and lymphocytes given their high abundance in the meninges as well as emerging evidence suggesting important roles for these immune cell lineages in TBI pathogenesis. However, it should be noted that there are many other populations of cells present in the meninges in which we were not able to adequately assess differential gene expres-sion due to their small population sizes. In future studies, it will be important to further interrogate the functions of these less abundant cell populations in TBI. Notably, the presence of plasmacytoid dendritic cells within the meninges is intriguing as they are known to be potent producers of type I IFNs (*Fitzgerald-Bocarsly et al., 2008*). It is possible that plasmacytoid cells could be contributing to the type I IFN signature seen with aging and TBI in the meninges; however, future work is needed to formally investigate this.

It is also known that sex differences play a role in outcomes after TBI both in human and animal models (*Gupte et al., 2019*). While men have a higher likelihood of sustaining a TBI, women have a higher likelihood of suffering worse outcomes (*Gupte et al., 2019*). Due to the number of other variables we were already considering for this study (age, time, and injury status), we were unable to include sex as a variable for our sequencing data. This leaves further opportunities for investiga-tion into how sex impacts the meningeal transcriptome in the context of both TBI and aging. High throughput sequencing techniques provide unique opportunities to understanding how sex affects CNS tissues at a cellular and transcriptional level.

While the meningeal immune response to head trauma in aged mice appears to be largely upreg-ulated, the meningeal immune response in young mice following injury appears to be held in check. One week following injury, although there is an upregulation in genes important for the inflammatory response in the macrophage and adaptive immune cell populations, there are other upregulated genes that are important for dampening this same immune response and promoting wound healing. For instance, the subsets of macrophages whose frequencies increase the most following injury are 'Anti-Inflammatory' and 'Resolution Phase' macrophages, both of which have been reported to exert wound-healing properties in injury models (*Russo et al., 2018*; *Stables et al., 2011*). Furthermore, while T cells were observed to upregulate genes associated with immune activation, survival, and adhesion, they also displayed increased expression of genes involved in dampening cytokine signaling and controlling the immune response in the meninges of young mice (e.g. *Cfl1*, *Socs2*, and *Cd52*). Moreover, the vast majority of these differentially expressed genes seen in young meninges at one week post-injury return to resting levels by 1.5 months following head trauma, suggesting a resolution of inflammation and a restoration of homeostasis in young mice. In contrast, aged mice do not appear to have this same success in resolving inflammatory responses following head trauma. In addition to the baseline inflammatory state of aged meninges that is characterized by increased expression of genes related to type I IFN signaling and antibody production, aged mice that received a TBI were found to further upregulate genes involved in driving inflammatory responses. Moreover, these injured aged mice were also shown to downregulate numerous genes involved in extracellular matrix reorganization and collagen production, which are two processes that are necessary for proper tissue

regeneration. These transcriptional alterations in aged meninges persist beyond a month following injury, with no indications of resolution.

It is well known that aged individuals have a higher morbidity and mortality than young individuals when experiencing a similar severity brain injury (*Susman et al., 2002*). The explanation for why the elderly experience these poorer outcomes following TBI is likely complex and multifaceted. Many studies have highlighted baseline changes in the aged brain that has been speculated to prime the elderly for differential responses following injury, including changes in the BBB, microglial dysfunction, and an overall increase in neuroinflammation (*Androvic et al., 2020*; *Yang et al., 2020*; *Marschallinger et al., 2020*). Furthermore, other findings support changes in the response to injury in the aged brain, including alterations in the type and number of immune cells recruited to the injury site, further increases in inflammatory gene signatures in the brain parenchyma, and elevated production of potentially neurotoxic molecules such as ROS and type I IFNs (*Chou et al., 2018*; *Morganti et al., 2016*; *Ritzel et al., 2018*; *Androvic et al., 2020*; *Ritzel et al., 2019*; *Kumar et al., 2013*; *Krukowski et al., 2018*; *Karve et al., 2016*; *Barrett et al., 2020*; *Baruch et al., 2014*; *Abdullah et al., 2018*; *Zhang et al., 2017*; *Itoh et al., 2013*). Our findings indicate that the meninges may also play a role in this differential response to head trauma seen in aging. In particular, it is possible that the increased baseline type I IFN gene signature and antibody production observed in aging potentially renders the aged brain prone to more severe clinical outcomes post-TBI.

Recent studies have implicated the meningeal lymphatic system, which resides in the dura, in modulating inflammation in the brain following TBI and sub-arachnoid hemorrhage (*Bolte et al., 2020*; *Chen et al., 2020*; *Pu et al., 2019*). In these studies, impairments in the meningeal lymphatic system prior to brain injury were found to result in increased gliosis and worsened behavioral outcomes (*Bolte et al., 2020*; *Chen et al., 2020*). Interestingly, the meningeal lymphatic system is also known to be impaired in aging (*Da Mesquita et al., 2018*; *Ahn et al., 2019*; *Ma et al., 2017*), and we have previously shown that the rejuvenation of the meningeal lymphatic vasculature in aged mice dampens the subsequent gliosis following TBI (*Bolte et al., 2020*). How the meningeal lymphatic system might modulate meningeal immunity before and after injury remains to be investigated. Furthermore, whether the meningeal lymphatic impairment in aging contributes to the overall increase in inflammation seen in the aged meninges is another area for future investigation.

## Conclusions

Overall, the findings presented here provide new insights into the meningeal response to brain injury and aging. We show that TBI results in broad gene expression changes in discrete cell populations following injury in young mice. Specifically, we demonstrate that there is an increase in the frequency of fibroblasts and macrophages 1 week following injury in young mice. Furthermore, we provide evidence that the transcriptional environment in the aged meninges is drastically altered. At baseline, the aged meninges show increases in gene expression patterns associated with type I IFN signaling and antibody production by B cells. However, upon injury, the aged meninges further upregulate genes involved in immune system activation, while downregulating genes critical for tissue remodeling. Improved understanding of how the meninges respond to brain injury in youth and aging will help shed light on why the elderly have poor outcomes following TBI and may help to identify opportunities for targeted therapies to improve outcomes following TBI.

# Materials and methods

**Key resources table**

| Reagent type (species) or resource | Designation | Source or reference | Identifiers | Additional information |
|---|---|---|---|---|
| Antibody | Anti-CD45.2 EF450 (rat monoclonal) | Thermo Scientific | Catalog # 11-0451-82, Clone 30-F11 | Flow(1:200) |
| Antibody | Anti-B220 PE-Cy7 (rat monoclonal) | BioLegend | Catalog # 103222, Clone RA3-6B2 | Flow(1:200) |
| Antibody | Anti-CD19 FITC (rat monoclonal) | eBioscience | Catalog # 11-0193-81, Clone eBio1D3 | Flow(1:200) |

*Continued on next page*

*Continued*

| Reagent type (species) or resource | Designation | Source or reference | Identifiers | Additional information |
|---|---|---|---|---|
| Antibody | Anti-J chain (rabbit monoclonal) | Invitrogen | Catalog # MA5-16419, Clone: SP105 | IF(1:200) |
| Antibody | Anti-Collagen I (rabbit polyclonal) | Abcam | Catalog # ab21286 | IF(1:200) |
| Antibody | Anti-Lyve-1-Alexa Fluor 488 (rat monoclonal) | eBioscience | Catalog # 53-0443-82, Clone ALY7 | IF(1:200) |
| Antibody | Anti-Iba1 (goat polyclonal) | Abcam | Catalog # ab5076 | IF(1:300) |
| Antibody | Anti-GFAP (rat monoclonal) | Thermo Fisher Scientific | Catalog # 13–0300, Clone 2.2B10 | IF(1:1000) |
| Antibody | Anti-MHC Class II 660 (rat monoclonal) | eBioscience | Catalog # 14-5321-82, Clone M5/114.12.2 | IF(1:100) |
| Antibody | Anti-CD31 (armenian hamster monoclonal) | Millipore Sigma | Catalog # MAB1398Z, Clone 2H8 | IF(1:200) |
| Antibody | Anti-B220 (rat monoclonal) | Thermo Fisher Scientific | Catalog # 14-0452-82, Clone RA3-6B2 | IF(1:200) |
| Antibody | Anti-NeuN (mouse monoclonal) | EMD Millipore | Catalog # Mab277, Clone A60 | IF(1:500) |
| Antibody | Donkey anti- rat Alexa Fluor 488 (donkey polyclonal) | Thermo Fisher Scientific | Catalog # A-21208 | IF(1:1000) |
| Antibody | Donkey anti- goat Alexa Fluor 647 (donkey polyclonal) | Thermo Fisher Scientific | Catalog # A-21447 | IF(1:1000) |
| Antibody | Donkey anti-rat Alexa Fluor 594 (donkey polyclonal) | Thermo Fisher Scientific | Catalog # A-21209 | IF(1:1000) |
| Antibody | Donkey anti-rabbit Alexa Fluor 647 (donkey polyclonal) | Thermo Fisher Scientific | Catalog # A-31573 | IF(1:1000) |
| Antibody | Alexa Fluor 488 anti-Armenian Hamster (goat polyclonal) | Jackson ImmunoResearch | Catalog # 127-545-160, RRID: AB_2338997 | IF(1:1000) |
| Other | Fixable Viability Dye eFlour 506 | eBioscience | Catalog # 65-0866-18 | Flow(1:800) |
| Other | Absolute counting beads | Life Technologies | Catalog # C36950 | Used for counting cells in flow cytometry. See '*Flow cytometry*'. |
| Other | Prolong Gold antifade reagent | Invitrogen | Catalog # P36930 | Used for mounting tissues for confocal imaging. See '*Immunohistochemistry, imaging, and quantification*'. |
| Commercial assay, kit | RNeasy Micro Kit | Qiagen | Catalog # 74004 | |
| Commercial assay, kit | SensiFAST cDNA synthesis kit | Bioline | Catalog # BIO-65054 | |
| Commercial assay, kit | SensiFAST Probe No-ROX Kit | Bioline | Catalog # BIO-86005 | |
| Sequence-based reagent | *Gapdh* | Life Technologies | Catalog # 4331182 | Assay ID Mm99999915_g1 |
| Sequence-based reagent | *Ifnar1* | Life Technologies | Catalog # 4331182 | Assay ID Mm00439544_m1 |
| Sequence-based reagent | *Irf5* | Life Technologies | Catalog # 4331182 | Assay ID Mm00496477_m1 |
| Sequence-based reagent | *Ifnb1* | Life Technologies | Catalog # 4331182 | Assay ID Mm00439552_s1 |
| Sequence-based reagent | *Ifi203* | Life Technologies | Catalog # 4331182 | Assay ID Mm00492601_m1 |
| Sequence-based reagent | *Col4a1* | Life Technologies | Catalog # 4331182 | Assay ID Mm01210125_m1 |

*Continued on next page*

*Continued*

| Reagent type (species) or resource | Designation | Source or reference | Identifiers | Additional information |
|---|---|---|---|---|
| Sequence-based reagent | *Col5a2* | Life Technologies | Catalog # 4331182 | Assay ID Mm01254423_m1 |
| Other | Collagenase VIII | Sigma Aldrich | Catalog # 9001-12-1 | Used for meningeal digestion for flow cytometry. See '*Meningeal tissue collection*'. |
| Other | Collagenase D | Sigma Aldrich | Catalog # 11088866001 | Used for meningeal digestion for flow cytometry. See '*Meningeal tissue collection*'. |
| Other | DAPI | Sigma Aldrich | Catalog # D9542 | IF(1:1000), See '*Immunohistochemistry, imaging, and quantification*'. |

## Mice

All mouse experiments were performed in accordance with the relevant guidelines and regulations of the University of Virginia and were approved by the University of Virginia Animal Care and Use Committee. Young C57BL/6J wild-type (WT) mice ranging from 8 to 12 weeks of age (RRID: IMSR_JAX:000664) were obtained from Jackson Laboratories. All WT aged mice were approximately 20 months of age and were obtained from the National Institute on Aging (NIA) Aged Rodent Colonies. The mice from the NIA Aged Rodent Colonies were originally derived from the Jackson colonies. Upon arrival, aged mice were housed in University of Virginia facilities for at least 2 months before use. Other young mice ordered directly from the Jackson Laboratories were housed in the local University of Virginia facility for at least 2 weeks before use. Mice were housed in specific pathogen-free conditions under standard 12 hr light/dark cycle conditions in rooms equipped with control for temperature (21 ± 1.5°C) and humidity (50 ± 10%). Male mice were used for all studies.

## Traumatic brain injury

This injury paradigm was adapted from the published Hit and Run model (*Ren et al., 2013*; *Bolte et al., 2020*). Mice were anesthetized by 4% isoflurane with 0.3 kPa $O_2$ for 2 min and then the right preauricular area was shaved. The mouse was placed prone on an 8x4 x 4-inch foam bed (type E bedding, open-cell flexible polyurethane foam with a density of approximately 0.86 pounds per cubic feet and a spring constant of approximately 4.0 Newtons per meter) with its nose in a nosecone delivering 1.5% isoflurane (purchased from Foam to Size, Ashland VA). The head was otherwise unsecured. The device used to deliver TBI was a Controlled Cortical Impact Device (Leica Biosystems, 39463920). A 3 mm impact probe was attached to the impactor device which was secured to a stereotaxic frame and positioned at 45 degrees from vertical. In this study, we used a strike depth of 2 mm, 0.1 s of contact time and an impact velocity of 5.2 meters (m) per second (s). The impactor was positioned at the posterior corner of the eye, moved 3 mm towards the ear and adjusted to the specified depth using the stereotaxic frame. A cotton swab was used to apply water to the injury site and the tail in order to establish contact sensing. To induce TBI, the impactor was retracted and dispensed once correctly positioned. The impact was delivered to the right inferior temporal lobe of the brain. Following impact, the mouse was placed supine on a heating pad and allowed to regain consciousness. After anesthesia induction, the delivery of the injuries took less than 1 minute. The time until the mouse returned to the prone position was recorded as the righting time. Upon resuming the prone position, mice were returned to their home cages to recover on a heating pad for six hours with soft food. For Sham procedures, mice were anesthetized by 4% isoflurane with 0.3 kPa $O_2$ for 2 min and then the right preauricular area was shaved. The mouse was placed prone on a foam bed with its nose secured in a nosecone delivering 1.5% isoflurane. The impactor was positioned at the posterior corner of the eye, moved 3 mm towards the ear and adjusted to the specified depth using the stereotaxic frame. A cotton swab was used to apply water to the injury site and the tail in order to establish contact sensing. Then, the impactor was adjusted to a height where no impact would occur, and was retracted and dispensed. Following the Sham procedure, the mouse was placed supine on a heating pad and allowed to regain consciousness. Mice were allowed to recover on the heating pad in their home cages for 6 hr with soft food before being returned to the housing facilities.

## Meningeal tissue collection

Mice were euthanized with $CO_2$ and then transcardially perfused with 20 mL 1 x PBS. The skin and muscle were stripped from the outer skull and the skullcap and the adherent meninges dorsal to the ear canal were collected using surgical scissors. Dorsal meningeal collection did not include the direct site impacted by the TBI. After further processing, as detailed below for each individual experiment, meningeal dissection from the skullcap was done under an S6D stereomicroscope (Leica) using Dumont #5 forceps (Fine Science Tools).

- For meningeal whole mount preparation, the skullcaps were fixed in 4% PFA for 6 hr at room temperature. Then the meninges (dura mater and some arachnoid mater) were carefully dissected in 1 x PBS and placed in 0.05% sodium azide in PBS at 4 °C until further use.
- For meninges collection for scRNA-seq, the skullcap was removed and placed into DMEM medium. Meninges were then scraped from the skullcap in 1 x PBS and processed further to create a single cell suspension as described in the method's section entitled 'Meningeal preparation for scRNA-seq' below.
- For meninges collection for bulk RNA-seq, the skullcap was removed and placed into DMEM medium. The meninges were scraped from the skullcap in 1 x PBS and were immediately snap-frozen at −80 °C in TRIzol (15596018, Life Technologies), until further use.
- For meninges collection for flow cytometry, the skullcap was removed and the meninges were dissected from the skullcap in 1 x PBS. The tissue was then digested in 1 X HBSS (without Ca2 +or Mg2+Gibco) with Collagenase D (1 mg/mL, Sigma-Aldrich) and Collagenase VIII (1 mg/mL, Sigma Aldrich) at 37 °C for 30 minutes, before being passed through a 70 μm strainer (Corning). Cells were pelleted, resuspended, and kept on ice.

## Other tissue collection

Mice were euthanized with $CO_2$ and then transcardially perfused with 20 mL 1 x PBS. Spleens for flow cytometry were harvested, placed into cold cRPMI, then mechanically homogenized and washed through a 40 μm strainer (Corning). Cells were resuspended in RBC lysis solution (0.16 M $NH_4Cl$) for 2 min. Cells were then washed, resuspended, and kept on ice until acquisition. Brains for immunohistochemistry were removed and kept in 4% PFA for 24 hr and then cryoprotected with 30% sucrose for 3 days. A 4 mm coronal section of brain tissue that surrounded the site of injury was removed using a brain sectioning device and then frozen in Tissue-Plus OCT compound (Thermo Fisher Scientific). Fixed and frozen brains were sliced (50 μm thick sections) with a cryostat (Leica) and kept in PBS + 0.05% sodium azide in PBS at 4 °C until further use.

## Flow cytometry

Meningeal and spleen single-cell suspensions were pipetted into a 96-well plate and pelleted. Cells were treated with 50 μL of Fc block (0.1% rat gamma globulin [Jackson ImmunoResearch], 1 μg/mL of 2.4G2 [BioXCell]) for 10 min at room temperature. Cells were then stained for surface markers and incubated with a fixable live/dead viability dye for 30 min at 4 °C using the following surface markers at a concentration of 1:200: CD45.2 EF450 (30-F11, Thermo Scientific), B220 PE-Cy7 (103222, BioLegend), and CD19 FITC (11-0193-81, eBioscience). Fixable Viability Dye eFluor 506 (65-0866-18, eBioscience) live/dead dye was used at a 1:800 dilution. Splenocytes were stained with the live/dead viability dye and were used for compensation controls. Finally, samples were resuspended in FACS buffer and acquired on a Gallios flow cytometer (Beckman Coulter). Cell counts were determined using absolute counting beads (Life Technologies, C36950) pipetted into samples just prior to acquisition. Data analysis was performed with FlowJo software v.10.

## RNA extraction and sequencing

For RNA extraction from the meningeal tissue, the meninges were harvested as described in the 'Tissue Collection' methods section above and snap-frozen in 500 μL TRIzol Reagent (15596018, Life Technologies) and stored at −80 °C until further use. For each of the four experimental groups (Young Sham, Aged Sham, Young TBI and Aged TBI) 2 dorsal meningeal samples were combined to create 1 biological replicate. Three biological replicates were used for each experimental group yielding a total of 12 samples comprised of 2 meninges each. After defrosting on ice, 10 silica beads were added to each tube and the tissue was homogenized for 30 s using a mini bead beater. Following

homogenization, the samples were centrifuged for 12,000 xg for 10 min at 4 °C. The supernatant was transferred to a new tube and incubated at room temperature for 5 min. Next, 0.1 mL of chloroform was added to the supernatant, vortexed, incubated for 2 min at room temperature and then centrifuged at 12,000 xg for 15 min at 4 °C. The top aqueous phase was transferred into a new Eppendorf tube and the RNeasy Micro Kit (74004, Qiagen) was used to isolate the RNA. The quality and concentration of the RNA was assessed using a plate reader. RNA was frozen at −80 °C until sent for sequencing. For sequencing, total RNA samples were sent to GENEWIZ for library preparation and paired-end sequencing. Sequencing was ordered for 20–30 million reads per sample.

### cDNA synthesis and qPCR

Meninges were harvested and RNA was isolated as described in 'RNA extraction and sequencing'. cDNA synthesis was performed using a SensiFAST cDNA synthesis kit (BIO-65054, Bioline) and cDNA was diluted to a concentration of 3 ng/µl in DNAse/RNAse free water. SensiFAST Probe No-ROX Kit (BIO-86005, Bioline) was used in combination with primers for *Gapdh, Ifnar1, Irf5, Ifnb1, Ifi203, Col4a1* and *Col5a2* (4331182, Life Technologies) to perform qPCR.

### Meningeal preparation for scRNA-seq

The day before meningeal harvest, Eppendorf tubes were coated with FACS buffer (1% BSA, 1 mM EDTA in PBS) overnight. Mice were euthanized with $CO_2$ and then transcardially perfused with ice-cold PBS with heparin (0.025%). The skull caps were prepared as described in 'Meningeal tissue collection'. Meninges were peeled from the skull cap and placed in ice-cold DMEM for the entirety of collection. The meninges from 5 mice that had received TBI 1 week prior were pooled as one biological replicate. The meninges from 5 mice that had received a Sham procedure 1 week prior were pooled as one biological replicate. These 2 samples were then processed for scRNA sequencing. Meninges were then digested for 15 min at 37 °C with constant agitation using 1 mL of pre-warmed digestion buffer (DMEM, with 2% FBS, 1 mg/mL collagenase VIII (Sigma Aldrich), and 0.5 mg/mL DNase I (Sigma Aldrich)). The enzymes were neutralized with 1 mL of complete medium (DMEM with 10% FBS) and meninges were then filtered through a 70 µm cell strainer. An additional 2 mL of FACS buffer was added, samples were centrifuged at 400 xg for 5 min, and samples were resuspended in FACS buffer. After 2 washes, cells were resuspended in FACS buffer with DAPI (0.2 µg/mL). Singlet gates were selected using pulse width of the side scatter and forward scatter. Live cells were selected based on the lack of DAPI staining. Cells were sorted into 1.5 mL tubes with ice cold DMEM. Following sorting, cells were centrifuged again at 450 xg for 4 min and the media was aspirated. Cells were resuspended in 200 µL 0.04% BSA in PBS (0.04% non-acetylated BSA) and centrifuged again. Cells were counted in 20 µL of 0.04% BSA in PBS using trypan blue. Approximately 4000 cells per sample were loaded onto a 10 X Genomics Chromium platform to generate cDNAs carrying cell- and transcript-specific barcodes and sequencing libraries constructed using the Chromium Single Cell 3' Library & Gel Bead Kit 2. Libraries were sequenced on the Illumina NextSeq using pair-ended sequencing, resulting in approximately 50,000 reads per cell. The sequencing mode was as follows: Read1 (28 bp)+Index (8 bp)+Read2 (98 bp).

### scRNA-seq analysis

The raw sequencing reads (FASTQ files) were aligned to the Genome Research Consortium (GRC) mm10 mouse genome build using Cell Ranger (v1.3.1) which performs alignment, filtering, barcode counting and unique molecular identifier (UMI) counting. RStudio (v1.2.5033) was used for all downstream analyses and Seurat (v.3.9.9) was used for filtering out low-quality cells, normalization of the data, determination of cluster defining markers and graphing of the data on UMAP (*Stuart et al., 2019*; *Butler et al., 2018*). Only one sequencing run was performed therefore there was no need for batch correction. Initially, there were 2261 cells collected from the Sham mice and 4022 cells collected from the TBI mice. Low-quality cells were excluded in an initial quality-control (QC) step by removing cells with fewer than 150 unique genes and cells expressing more than 5,000 unique genes in effort to remove doublets and triplets (Sham total: 2257, TBI total: 4018). Cells with transcriptomes that were more than 20% mitochondrial-derived were removed and cells with more than 5% hemoglobin among their expressed genes were also removed (Sham total: 2049, TBI total: 3775). Using Seurat, genes with high variance were selected using the FindVariableGenes() function, then the dimensionality of

the data was reduced by principle component analysis (PCA) and identified by random sampling of 20 significant principal components (PCs) for each sample with the PCElbowPlot() function. Cells were clustered with Seurat's FindClusters() function. Absolute cell counts for each population can be found in *Table 1*. scCATCH (v2.1) was used for automated cluster naming (*Shao et al., 2020*), and all cluster names were manually checked due to the lack of literature regarding meningeal cell populations. Next, differential gene expression analysis was performed within the clusters using the ZINB-WaVE (v1.12.0) and DESeq2 (v1.30.0) packages (*Risso et al., 2018*). Cytoscape (v3.8.0) and ToppCluster (https://toppcluster.cchmc.org/) were used for network analyses (*Shannon et al., 2003*; *Chen et al., 2007*). Data was organized and graphs were created using ggplot2, tidyverse, treemapify, circlize, Seurat and dplyr (*Wickham, 2016*; *Wickham et al., 2019*). Pseudotime analysis was conducted using Monocle3 (v0.2.3.0) (*Trapnell et al., 2014*). NicheNet was used to analyze predicted cellular interactions within the dataset (*Browaeys et al., 2020*).

## Bulk RNA-seq analysis

The raw sequencing reads (FASTQ files) were aligned to the GRC mm10 mouse genome build using the splice-aware read aligner HISAT2 (*Kim et al., 2019*). The data was cleaned by removing reads that aligned to uninformative regions of the genome and PCR duplicates. The proportion of PCR duplicates, proportion of reads that align with genes, and proportion of uninformative reads were analyzed for each sample to ensure there were no samples that were outliers regarding data quality. Samtools was used for quality control filtering (*Li et al., 2009*). Reads were sorted into feature counts with HTSeq (*Anders et al., 2015*). DESeq2 (v1.30.0) was used to normalize the raw counts based on read depth, perform principal component analysis, and conduct differential expression analysis (*Love et al., 2014*). For assessing differential gene expression in this dataset, the 'categorize.deseq.df' function was used to classify genes in to 'activated', 'repressed', and 'not different' categories as defined by an adjusted p-value of less than 0.1. The p-values were corrected with the Benjamini-Hochberg procedure to limit false positives arising from multiple testing. The gene set collections from MSigDB were used for differential gene set enrichment analysis (*Liberzon et al., 2015*). The analysis itself was performed using the Seq2Pathway, fgsea, tidyverse, and dplyr software packages. Heatmaps were generated using the pheatmap R package while other plots were made with the lattice or ggplot2 packages.

## Immunohistochemistry, imaging, and quantification

For immunofluorescence staining, meningeal whole mounts and floating brain sections in PBS and 0.05% sodium azide were blocked with a solution containing 2% goat serum or 2% donkey serum, 1% bovine serum albumin, 0.1% triton, 0.05% tween-20, and 0.05% sodium azide in PBS for 1.5 hr at room temperature (RT). This blocking step was followed by incubation with appropriate dilutions of primary antibodies in blocking solution at 4 °C overnight. The primary antibodies and their dilutions include: anti-Collagen I (Abcam, ab21286, 1:200), anti-J chain (Invitrogen, SP105, 1:200), anti-Lyve-1-Alexa Fluor 488 (eBioscience, clone ALY7, 1:200), anti-Iba1 (Abcam, ab5076, 1:300), anti-GFAP (Thermo Fisher Scientific, 2.2B10, 1:1000), anti-MHC Class II 660 (eBioscience, M5/114.15.2, 1:100), anti-CD31 (Millipore Sigma, MAB1398Z, clone 2H8, 1:200), anti-B220 (Thermo Fisher Scientific, RA3-6B2, 1:200) and anti-NeuN (EMD Millipore, Mab277, clone A60, 1:500). Meningeal whole mounts and brain sections were then washed three times for 10 minutes at room temperature in PBS and 0.05% tween-20, followed by incubation with the appropriate goat or donkey Alexa Fluor 488, 594 or 647 anti-rat, -goat or -rabbit (Thermo Fisher Scientific, 1:1000) or -Armenian hamster (Jackson ImmunoResearch, 1:1000) IgG antibodies for 2 hr at RT in blocking solution. The whole mounts and brain sections were then washed 3 times for 10 min at RT before incubation for 10 min with 1:1000 DAPI in PBS. The tissue was then transferred to PBS and mounted with ProLong Gold antifade reagent (Invitrogen, P36930) on glass slides with coverslips.

Slide preparations were stored at 4 °C and imaged using a Lecia TCS SP8 confocal microscope and LAS AF software (Leica Microsystems) within one week of staining. Quantitative analysis of the acquired images was performed using Fiji software. Imaging parameters for brightness, contrast, and threshold values were applied uniformly throughout each experiment. Additionally, tears in the meninges were excluded when performing the analyses. For the assessment of collagen expression and J-chain quantification, 5 meningeal whole mounts were included per experimental group. For

collagen, 10 x images were taken of meningeal whole mounts. To quantify the collagen, the corrected total cellular fluorescence (CTCF) was used, which takes into account the area of the meninges, the average fluorescent intensity of that area, and the fluorescent intensity of the background, as given by the formula: Mean fluorescence of meningeal whole mounts - (area of meningeal whole mount x mean fluorescence of background). For J-chain quantification, a uniformly sized 20 x image of the superior sagittal sinus was taken for each sample with 5 samples in each experimental group. The number of J-chain puncta was then quantified (puncta threshold: 5–10 microns) using the 'Analyze Particle' tool. For quantification of B220 + cells, 20 x scans (n=5 per group) were taken of the entire transverse sinus. The number of B220 + cells were manually counted and quantified by a blinded experimenter. For CD31 staining, percent area was quantified from whole mount meninges scanned at 10 x. For MHC II staining, eight meningeal whole mounts per experimental group were imaged at 10 x. The number of MHC II puncta was quantified.

For the assessment of gliosis in the injured and uninjured brains in *Figure 1—figure supplement 1*, two representative brain sections from the site of the lesion (approximately −0.74–0 bregma) or the corresponding area in Sham animals were fully imaged and at least four animals were included per experimental group. The full brain section was adjusted for brightness and contrast uniformly for each experiment and each hemisphere was traced, and then the threshold was uniformly set for each experiment to select for stained cells. The percent area of coverage of each immunohistochemical markers was calculated for the hemisphere ipsilateral to the injury site (right) for each brain section. The mean percent area fraction was calculated using Microsoft Excel. For *Figure 1—figure supplement 1*, high magnification images (63 x) were taken directly adjacent to the site of the injury.

## Statistical analysis and reproducibility

Sample sizes were chosen on the basis of standard power calculations (with $\alpha$=0.05 and power of 0.8). Experimenters were blinded to the identity of experimental groups from the time of euthanasia until the end of data collection and analysis. Statistical analysis was performed using RStudio (v1.2.5033) and GraphPad Prism (v8.4.3). Individual statistical analyses for each experiment are detailed in the corresponding figure legends.

## Availability of data and material

All data and genetic material used for this paper are available in the GEO repository under accession number GSE206941. All code used for analysis is available at (https://github.com/danielshapiro1/MeningealTransciptome, copy archived at swh:1:rev:d75b20d74f147524296630b9ec7a1c9c5a9f124f; *Daniel, 2022*).

## Acknowledgements

We thank members of the Lukens lab and the Center for Brain Immunology and Glia (BIG) for valuable discussions. Graphical illustrations in *Figure 1*, *Figure 5*, and *Figure 1—figure supplement 1* were made using BioRender (https://biorender.com/). This work was supported by The National Institutes of Health/National Institute of Neurological Disorders and Stroke (R01NS106383; awarded to JRL), The Alzheimer's Association (AARG-18–566113; awarded to JRL), The Owens Family Foundation (Awarded to JRL), and The University of Virginia Research and Development Award (Awarded to JRL). ACB, ABD, MAK and WFM were supported by a Medical Scientist Training Program Grant (5T32GM007267-38). ACB was supported by an Immunology Training Grant (5T32AI007496-25), a Wagner Fellowship, and the National Institute on Aging (NIA, F30AG069396-01). ABD was supported by a Biomedical Data Sciences Training Grant (T32LM012416). MAK was supported by the National Institute of Allergy and Infectious Diseases (NIAID, F30AI154740). HEE was supported by a Wagner Fellowship.

## Additional information

### Funding

| Funder | Grant reference number | Author |
|---|---|---|
| National Institute of Neurological Disorders and Stroke | R01NS106383 | John R Lukens |
| Alzheimer's Association | AARG-18-566113 | John R Lukens |
| The Owens Family Foundation | Pilot Grant | John R Lukens |
| National Institutes of Health | 5T32GM007267-38 | Ashley C Bolte Wei Feng Ma Michael A Kovacs Arun B Dutta |
| National Institutes of Health | 5T32AI007496-25 | Ashley C Bolte |
| National Institute on Aging | F30AG069396-01 | Ashley C Bolte |
| National Institutes of Health | T32LM012416 | Arun B Dutta |
| National Institute of Allergy and Infectious Diseases | F30AI154740 | Michael A Kovacs |
| University of Virginia | Wagner Fellowship | Hannah E Ennerfelt Ashley C Bolte |
| University of Virginia | Research and Development Award | John R Lukens |

The funders had no role in study design, data collection and interpretation, or the decision to submit the work for publication.

### Author contributions

Ashley C Bolte, Conceptualization, Data curation, Formal analysis, Supervision, Funding acquisition, Validation, Investigation, Visualization, Methodology, Writing – original draft, Writing – review and editing; Daniel A Shapiro, Data curation, Formal analysis, Investigation, Visualization, Methodology, Writing – review and editing; Arun B Dutta, Wei Feng Ma, Data curation, Formal analysis, Visualization, Methodology; Katherine R Bruch, Data curation, Validation, Visualization; Michael A Kovacs, Hannah E Ennerfelt, Conceptualization, Data curation; Ana Royo Marco, Supervision, Methodology; John R Lukens, Conceptualization, Resources, Data curation, Supervision, Funding acquisition, Investigation, Methodology, Project administration, Writing – review and editing

### Author ORCIDs

Ashley C Bolte (ID) http://orcid.org/0000-0003-4013-3759
Daniel A Shapiro (ID) http://orcid.org/0000-0002-1289-8036
Michael A Kovacs (ID) http://orcid.org/0000-0002-4298-9609
Ana Royo Marco (ID) http://orcid.org/0000-0002-2801-2746
John R Lukens (ID) http://orcid.org/0000-0002-6795-0866

### Ethics

This study was performed in strict accordance with the recommendations in the Guide for the Care and Use of Laboratory Animals of the National Institutes of Health. All mouse experiments were performed in accordance with the relevant guidelines and regulations of the University of Virginia and were approved by the University of Virginia Animal Care and Use Committee (University of Virginia animal protocol #4064).

### Decision letter and Author response

Decision letter https://doi.org/10.7554/eLife.81154.sa1
Author response https://doi.org/10.7554/eLife.81154.sa2

## Additional files

### Supplementary files
• MDAR checklist

### Data availability

Sequencing data have been deposited in GEO under accession code GSE206941. All code used for analysis is available at https://github.com/danielshapiro1/MeningealTransciptome (copy archived at swh:1:rev:d75b20d74f147524296630b9ec7a1c9c5a9f124f). All data generated or analyzed during this study are included in the manuscript and supporting file; Source Data files have been provided for all figures.

The following dataset was generated:

| Author(s) | Year | Dataset title | Dataset URL | Database and Identifier |
|---|---|---|---|---|
| Bolte AC, Shapiro DA, Dutta AB, Bruch KR, Marco AR, Lukens JR, Ma WF | 2022 | The meningeal transcriptional response to traumatic brain injury and aging | http://www.ncbi. nlm.nih.gov/geo/ query.acc.cgi?acc= GSE206941 | NCBI Gene Expression Omnibus, GSE206941 |

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
