## [Editor Report]

The authors provide single RNA-seq analysis of traumatic brain injury (TBI) that particularly addresses the question of why older individuals may have poor recovery. Compelling complementary and high-end approaches are taken to demonstrate the long-lasting effects that TBI drives in the brain. This important manuscript will be of interest to readers in the field(s) of neuroimmunology, aging, and traumatic brain injury.

---

## [Decision Letter]

**Decision letter after peer review:**

Thank you for submitting your article "The meningeal transcriptional response to traumatic brain injury and aging" for consideration by *eLife*. Your article has been reviewed by 2 peer reviewers, and the evaluation has been overseen by a Reviewing Editor and Christian Büchel as the Senior Editor. The following individuals involved in the review of your submission have agreed to reveal their identity: Brian S Kim (Reviewer #2).

Essential revisions:

Revisions focusing on the improvement of bioinformatic analysis along the points below will improve the manuscript

1) Better integration of scRNAseq and bulkseq to determine whether these changes in the subacute timepoint carryover chronically.

2) Bulkseq data could be better presented for clarity and show unbiased examinations of the bulkseq data first (e.g. pathway, ontologies, or gsea, etc.), Define which genes(ets)/ontologies are uniquely exaggerated due to the Aged TBI conditions.

3) Apply NichNet to probe putative intercellular communication networks that would lend insight into whether there are macrophage-fibroblast or macrophage-B cell interactions that drive non-resolving alterations.

*Reviewer #1 (Recommendations for the authors):*

Overall, this is a strong manuscript and valuable resource supporting the chronic contributions of inflammatory response in the aged brain following TBI. The focus on the neuroimmune response of the meningeal compartment is important and the datasets (scRNAseq and bulk) will provide a valuable resource for the research community that focuses on these research areas. While enthusiasm for the majority of the work is strong, there are several areas that could be bolstered to better support the Authors' endeavors.

Cell class-level DEGs derived from scRNAseq studies were not well integrated/quantified from the bulkseq data. Sup Figures4-6 have unique genesets, and/or Figures 2a, 3a, and 4a have calculated trauma-responsive genes, yet these were not integrated very well from the bulkseq studies to determine whether these changes in the subacute timepoint carryover chronically.

Bulkseq data could be better presented for clarity. Figure 7 seems to belong within/after Figure 5. Overall, it would be better presented to show unbiased examinations on the bulkseq data first (e.g. pathway, ontologies, or gsea, etc.), then show scRNseq-defined gene/cell class quantification (e.g. Fig6).

Figure 7a and Figure 5d don't capture which DEGs (up or down) are unique to the Aged TBI condition as the baseline is calculated from Young TBI, rather than Aged Sham, similarly Young Sham vs. Young TBI were not elucidated. There are several methods for distinguishing these contrasts (4-way Venn, Upset plot, etc.), which would more robustly define which genes(ets)/ontologies are uniquely exaggerated due to the Aged TBI condition.

*Reviewer #2 (Recommendations for the authors):*

Have the authors tried programs like NichNet to probe putative intercellular communication networks that would lend insight into whether there are macrophage-fibroblast or macrophage-B cell interactions that drive non-resolving alterations? Could authors comment on what they think is going on in terms of possible epigenomic alterations based on other works possibly with aging?

---

## [Author Response]

Essential revisions:Revisions focusing on the improvement of bioinformatic analysis along the points below will improve the manuscript1) Better integration of scRNAseq and bulkseq to determine whether these changes in the subacute timepoint carryover chronically.2) Bulkseq data could be better presented for clarity and show unbiased examinations of the bulkseq data first (e.g. pathway, ontologies, or gsea, etc.), Define which genes(ets)/ontologies are uniquely exaggerated due to the Aged TBI conditions.3) Apply NichNet to probe putative intercellular communication networks that would lend insight into whether there are macrophage-fibroblast or macrophage-B cell interactions that drive non-resolving alterations.

We thank the Editors for their valuable suggestions and continued interest in our work. We have focused on these essential revisions as discussed below.

Reviewer #1 (Recommendations for the authors):Overall, this is a strong manuscript and valuable resource supporting the chronic contributions of inflammatory response in the aged brain following TBI. The focus on the neuroimmune response of the meningeal compartment is important and the datasets (scRNAseq and bulk) will provide a valuable resource for the research community that focuses on these research areas. While enthusiasm for the majority of the work is strong, there are several areas that could be bolstered to better support the Authors' endeavors.

We were pleased that the Reviewer found our work to be strong and that it would be a valuable resource to the field. As the Reviewer suggested, we have provided additional analyses to further understand the link between the single cell and bulk RNA-seq datasets and have clarified the analysis of the bulk RNA sequencing according to the Reviewer’s comments. We thank the Reviewer for their valuable suggestions as we feel that they have greatly helped to improve this study.

Cell class-level DEGs derived from scRNAseq studies were not well integrated/quantified from the bulkseq data. Sup Figures4-6 have unique genesets, and/or Figures 2a, 3a, and 4a have calculated trauma-responsive genes, yet these were not integrated very well from the bulkseq studies to determine whether these changes in the subacute timepoint carryover chronically.

This is an excellent point that is brought forth by the Reviewer, and therefore we have performed new analyses to better understand how our scRNA-seq and bulk RNA-seq gene sets are related. This new data analysis now appears in Figure 8 of the revised manuscript. First, we compared the differentially expressed genes in the T cells, B cells, Fibroblasts and Macrophages in the scRNA-seq dataset with the differentially expressed genes in the Young Sham vs. Aged Sham comparison, to determine whether some of these genes were differentially expressed with aging alone (Figure 8a in the revised manuscript). While there were 139 shared differentially regulated genes, a majority of the differentially expressed genes in each dataset were not shared (Figure 8a in the revised manuscript). When we looked more closely at the shared downregulated genes, we found that many were important for wound healing and maintenance of connective tissue (Figure 8c in the revised manuscript). These data suggest that some of the downregulated genes important for wound healing in the subacute timepoint after TBI remain chronically downregulated in the aged meninges, further supporting the idea that aged meninges may be less able to respond to injury at baseline. Some of the upregulated genes that were shared at the subacute timepoint and chronically in aging were genes that contribute to abnormal immune cell physiology, innate immune response, and immune cell activation, again supporting the notion that aged meninges have a baseline activation of the immune system compared with young meninges (Figure 8c in the revised manuscript).

Looking more closely at the differentially regulated genes shared between the Young TBI vs. Aged TBI bulk RNA sequencing comparison and the T cells, B cells, Fibroblasts, and Macrophages of the scRNA-seq dataset, we found 119 genes in common (Figure 8b in the revised manuscript). Similar to aging alone, many of the common upregulated genes were related to abnormal immune cell activation, reflecting the chronically activated immune response that occurs after TBI in aging (Figure 8d in the revised manuscript). Of the shared downregulated genes, many contribute to cell adhesion and response to endoplasmic reticulum stress (Figure 8d in the revised manuscript). Altogether, while a majority of the genes that were differentially expressed in both the bulk RNA and scRNA sequencing datasets were not shared, the common genes reflect a pattern of abnormal immune cell activation and a defective response to healing as demonstrated by the downregulation of genes important for extracellular matrix repair and cellular adhesion.

Bulkseq data could be better presented for clarity. Figure 7 seems to belong within/after Figure 5. Overall, it would be better presented to show unbiased examinations on the bulkseq data first (e.g. pathway, ontologies, or gsea, etc.), then show scRNseq-defined gene/cell class quantification (e.g. Fig6).

We thank the Reviewer for bringing forth an area of the manuscript that was unclear and for this thoughtful suggestion. We have made efforts to clarify this portion of the manuscript. Figure 5 is the first introduction to the bulk sequencing dataset. We removed the Venn diagram in Figure 7 and instead included upset plots in Figure 5 to more clearly delineate the comparisons within the dataset and which genes were shared/unique to each comparison, as suggested by the Reviewer. Figure 5 in the revised manuscript now includes all general, unbiased expression data from this bulk sequencing experiment.

Next, in Figure 6 we highlight the gene expression changes that were seen with aging alone. Figure 6 starts with unbiased analysis of the Young Sham vs. Aged Sham comparison (top 20 upand down-regulated genes, GO terms) and then we more closely examine some of the genes that are important for contributing to these GO terms still within the bulk sequencing dataset (Figure 6c,h). We include additional immunohistochemistry and flow cytometry studies to expand on these points and to validate this dataset. Finally, in Figure 7, we discuss the changes that were unique to TBI in aging and not shared with the changes seen in aging alone. In Figure 7 we also start with unbiased analyses of the genes unique to TBI in aging with GO terms (Figure 7a,d) and again look at some of the specific genes that contribute to these GO signatures within the bulk sequencing dataset (Figure 7b,e). We have now included some validation of these genes by qPCR. We thank the Reviewer for these suggestions and we believe with these modifications, the presentation of the bulk RNA-seq data is clearer. We have added text to the manuscript to clarify these points further.

Figure 7a and Figure 5d don't capture which DEGs (up or down) are unique to the Aged TBI condition as the baseline is calculated from Young TBI, rather than Aged Sham, similarly Young Sham vs. Young TBI were not elucidated. There are several methods for distinguishing these contrasts (4-way Venn, Upset plot, etc.), which would more robustly define which genes(ets)/ontologies are uniquely exaggerated due to the Aged TBI condition.

We thank the Reviewer for this thoughtful advice. We have now portrayed the bulk sequencing data using upset plots to better convey which genes are unique to TBI or shared between TBI and aging. Figure 7 shows the data for the genes that are unique to the aged TBI comparison and not shared with the aging alone comparison. Given the addition of this new upset plot in the revised manuscript, we have removed the Venn Diagram from Figure 7.

Reviewer #2 (Recommendations for the authors):Have the authors tried programs like NichNet to probe putative intercellular communication networks that would lend insight into whether there are macrophage-fibroblast or macrophage-B cell interactions that drive non-resolving alterations? Could authors comment on what they think is going on in terms of possible epigenomic alterations based on other works possibly with aging?

We thank the Reviewer for their suggestion to use software like NicheNet to investigate the communication networks between cells that may shed light on ligand-receptor interactions. We have used NicheNet to examine how macrophage and fibroblast signaling in the meninges affects other meningeal cells, as both were large populations in the meninges that experienced significant changes post-TBI.

We first examined how the macrophage cell populations might influence the gene expression patterns of the other major cell populations in our scRNA-seq dataset (T cells, B cells, Dendritic Cells, NK cells, Fibroblasts, and Endothelial Cells) through inferred ligand-target interactions. This new data analysis is provided below for the Reviewer and also now appears in Figure 2—figure supplement 2 of the revised manuscript. *Tgfb1* was the top macrophage ligand that best predicted the gene expression patterns seen in these cell populations, indicating that the overall gene expression pattern represents one of cell growth, differentiation, and alternative macrophage activation (Figure 2—figure supplement 2a,b in the revised manuscript). Additionally, many of the top macrophage ligands were important for cell adhesion and phagocytosis signaling pathways, including *Itgam*, *Apoe*, *Vcam1*, *Selplg*, *Nectin1,* and *Itgb1*, all of which play critical roles in an inflammatory response (Figure 2—figure supplement 2a,b in the revised manuscript). Other top ligands instigate pro-inflammatory signaling pathways such as *Adam17*, which is involved in the processing of TNF at the surface of the cell, *Tnfsf13b*, which promotes activation and proliferation of B cells, and *C3*, which is part of the complement cascade (Figure 2—figure supplement 2a,b in the revised manuscript). When examining the predicted target genes that these top ligands would affect, they are very consistent with some of the gene expression patterns seen in the scRNAseq dataset. For example, signaling through *Tgfb1* is predicted to upregulate expression of multiple genes that affect collagen production including *Col1a1*, *Col1a2,* and *Col3a1* (Figure 2—figure supplement 2c in the revised manuscript), and we did see increased collagen production following TBI. Other ligands linked to heightening the inflammatory response such as *Adam17* and *Tnfsf13b* have predicted target genes including genes related to the B cell response (*Cd19, Cd79a,* and *Ighm*) and related to immune cell activation and antigen presentation (*Cd38* and *Cd40*) (Figure 2—figure supplement 2c in the revised manuscript). Finally, when we looked at how ligands from other cells in the meninges may impact the macrophage gene signature, we saw that ligands from multiple cell populations (Endothelial Cells, Dendritic Cells, T cells, and B cells) are likely responsible for shaping the macrophage gene signature (Figure 2—figure supplement 2d in the revised manuscript). This is not surprising given that monocytes and macrophages likely interact with all of the meningeal cell populations and exert a very important role in shaping the overall gene expression signatures seen after TBI. Overall, the predicted ligand-target interactions between the macrophage population and other major meningeal cell populations after brain injury illustrate a proliferative, pro-inflammatory state.